# Learning Bipartite Graphs: Heavy Tails and Multiple Components

**José Vinícius de M. Cardoso[1], Jiaxi Ying[2], Daniel P. Palomar[1,3]**
Department of Electronic and Computer Engineering[1]
Department of Mathematics[2]
Department of Industrial Engineering and Decision Analytics[3]
The Hong Kong University of Science and Technology
Clear Water Bay, Hong Kong SAR China
{jvdmc, jx.ying}@connect.ust.hk, palomar@ust.hk

## Abstract

We investigate the problem of learning an undirected, weighted bipartite graph under the Gaussian Markov random field model, for which we present an optimization formulation along with an efficient algorithm based on the projected gradient descent. Motivated by practical applications, where outliers or heavy-tailed events are present, we extend the proposed learning scheme to the case in which the data follow a multivariate Student-$t$ distribution. As a result, the optimization program is no longer convex, but a verifiably convergent iterative algorithm is proposed based on the majorization-minimization framework. Finally, we propose an efficient and provably convergent algorithm for learning $k$-component bipartite graphs that leverages rank constraints of the underlying graph Laplacian matrix. The proposed estimators outperform state-of-the-art methods for bipartite graph learning, as evidenced by real-world experiments using financial time series data.

## 1 Introduction

Efficient optimization formulations alongside scalable optimization algorithms have been critical tools for performing inference in graphical models. In particular, learning sparse, *unconstrained* Gaussian Markov random fields (MRF) has enabled a wide range of practical applications across a number of fields, including brain network analysis [1], single-cell sequencing [2], time-varying network estimation [3], time-series clustering [4] and model selection [5], and financial networks [6, 7]. Such an impact may be attributed to the early development of the graphical lasso algorithm via iterative block coordinate descent methods [8–10].

More recently, there has been a growing interest in *constraining* Gaussian MRFs into a particular family such as bipartite and $k$-component graphs for data clustering [11–14], and considering robust statistics that account for heavy-tailed events [15, 16]. In addition, constraints on the sign of the elements of the precision matrix of a Gaussian MRF have been employed to learn total positivity models [17–20] as well as Laplacian models in sparse settings [21–23]. Estimating undirected graphs under smooth signal assumptions has been investigated in [24–27].

In this paper, we consider the problem of learning undirected, weighted bipartite graphs under the MRF assumption. The key contributions of this paper include:

1. We design an efficient optimization algorithm based on the projected gradient descent (PGD) framework for the problem of learning an undirected, weighted bipartite graph from the Gaussian MRF framework.

36th Conference on Neural Information Processing Systems (NeurIPS 2022).

2. We extend our proposed formulation in order to accommodate the multivariate Student-$t$ distribution, which is often needed for real datasets that contain outliers or are inherently heavy-tailed. In this case, the optimization formulation turns out to be nonconvex. We design an optimization algorithm based on the majorization-minimization (MM) framework to obtain a stationary point of such problem.

3. We propose an efficient and provably convergent algorithm based on the alternating direction method of multipliers (ADMM) for learning $k$-component bipartite graphs that leverages the rank properties of the Laplacian matrix.

4. We present empirical results, using real-world datasets from the US stock market, which reveal that our formulations outperform state-of-the-art ones in terms of graph modularity and node label accuracy.

**Notation**: Matrices (vectors) are denoted by bold, italic, capital (lowercase) roman letters like $\boldsymbol{X}$, $\boldsymbol{x}$. Vectors are assumed to be column vectors. We use $\mathbf{1}_n$ (resp. $\boldsymbol{I}_n$) to denote the $n$-dimensional all-one vector (resp. identity matrix). The $(i, j)$ element of a matrix $\boldsymbol{X} \in \mathbb{R}^{n \times p}$ is denoted as $X_{ij}$. The $i$-th element of a vector $\boldsymbol{x}$ is denoted as $x_i$. The $i$-th row of $\boldsymbol{X}$ is denoted as $\boldsymbol{x}_i \in \mathbb{R}^{p \times 1}$. $\boldsymbol{X} \geq \boldsymbol{Y}$ implies that $X_{ij} \geq Y_{ij} \, \forall \, i, j$, whereas $\boldsymbol{X} \succeq \boldsymbol{Y}$ implies that $\boldsymbol{X} - \boldsymbol{Y}$ is positive semi-definite. For a square matrix $\boldsymbol{X}$, $\lambda_i (\boldsymbol{X})$ denotes the $i$-th smallest eigenvalue of $\boldsymbol{X}$. For a vector $\boldsymbol{x}$ and indices $j > i$, $\boldsymbol{x}_{i:j}$ denotes $(x_i, x_{i+1}, \ldots, x_j)^\top$. The operator $\mathsf{Diag} : \mathbb{R}^p \to \mathbb{R}^{p \times p}$ creates a diagonal matrix with the elements of an input vector along its diagonal. The operator $\mathsf{diag} : \mathbb{R}^{p \times p} \to \mathbb{R}^p$ extracts the diagonal of a square matrix. The optimal point of an optimization variable $\boldsymbol{v}$ is denoted as $\boldsymbol{v}^\star$.

## 2 Preliminaries & Related Works

An undirected, weighted, bipartite graph can be defined as a 4-tuple $\mathcal{G} \triangleq \{\mathcal{V}_r, \mathcal{V}_q, \mathcal{E}, \boldsymbol{W}\}$, where $\mathcal{V}_r \triangleq \{1, 2, \ldots, r\}$ and $\mathcal{V}_q \triangleq \{r + 1, r + 2, \ldots, r + q\}$ are the node sets associated to a group of objects and classes, respectively, where $p \triangleq r + q$ is the total number of nodes, and we assume that $r \gg q$, *i.e.*, there exist many more objects than classes; $\mathcal{E} \subseteq \{\{u, v\} : u \in \mathcal{V}_r, v \in \mathcal{V}_q\}$ is the edge set, that is, a subset of the set of all possible unordered pairs of nodes such that $\{u, v\} \in \mathcal{E}$ iff nodes $u$ and $v$ are connected; $\boldsymbol{W} \in \mathbb{R}_+^{p \times p}$ is the symmetric weighted adjacency matrix that satisfies $W_{ii} = 0, W_{ij} > 0$ iff $\{i, j\} \in \mathcal{E}$ and $W_{ij} = 0$, otherwise. The weighted Laplacian matrix of a graph is defined as $\boldsymbol{L} \triangleq \mathsf{Diag}(\boldsymbol{W}\mathbf{1}) - \boldsymbol{W}$.

Our goal is to learn a bipartite graph from data under probabilistic assumptions. Thus, the data generating process is initially assumed to be a zero-mean improper Gaussian Markov random field (IGMRF) [28] $\boldsymbol{x} \in \mathbb{R}^p$, such that $x_i$ is the random variable generating a signal measured at node $i$, whose rank-deficient precision matrix is modeled as a graph Laplacian matrix [21, 22, 29]. Assume we are given $n$ observations of $\boldsymbol{x}$, *i.e.*, $\boldsymbol{X} = [\boldsymbol{x}_1, \boldsymbol{x}_2, \ldots, \boldsymbol{x}_n]^\top$, $\boldsymbol{X} \in \mathbb{R}^{n \times p}$, $\boldsymbol{x}_i \in \mathbb{R}^{p \times 1}$. The goal of graph learning algorithms is to learn a Laplacian matrix given the data matrix $\boldsymbol{X}$.

To that end, the classical MLE of the Laplacian-constrained precision matrix of $\boldsymbol{x}$, on the basis of the observed data $\boldsymbol{X}$, may be formulated as the following optimization program [21, 22, 29]:

$$\underset{\boldsymbol{L} \succeq \mathbf{0}}{\mathsf{minimize}} \quad \mathrm{tr}\,(\boldsymbol{L}\boldsymbol{S}) - \log \det{}^* (\boldsymbol{L}), \text{ subject to } \boldsymbol{L}\mathbf{1} = \mathbf{0}, \ L_{ij} = L_{ji} \leq 0, \ i \neq j, \quad (1)$$

where $\boldsymbol{S}$ is a similarity matrix, *e.g.*, the sample covariance (or correlation) matrix $\boldsymbol{S} \propto \boldsymbol{X}^\top \boldsymbol{X}$, and $\det{}^*(\boldsymbol{L})$ is the pseudo-determinant of $\boldsymbol{L}$, *i.e.*, the product of its positive eigenvalues [30].

The Laplacian matrix $\boldsymbol{L}$ of a bipartite graph can be written as

$$\boldsymbol{L} = \begin{bmatrix} \mathsf{Diag}\,(\boldsymbol{B}\mathbf{1}_q) & -\boldsymbol{B} \\ -\boldsymbol{B}^\top & \mathsf{Diag}\,(\boldsymbol{B}^\top \mathbf{1}_r) \end{bmatrix}, \quad (2)$$

where $\boldsymbol{B} \in \mathbb{R}_+^{r \times q}$ contains the edge weights between the nodes of objects and the nodes of classes. Note that $\boldsymbol{L}$ satisfies $\boldsymbol{L}\mathbf{1} = \mathbf{0}$ and $L_{ij} = L_{ji} \leq 0 \, \forall \, i \neq j$ by definition.

In what follows, we briefly discuss the two main approaches proposed in the literature to tackle the estimation of bipartite graphs.

**Bipartite Structure.** The authors in [12] proposed the following optimization problem to learn a $k$-component bipartite graph from a given bipartite graph weights $\boldsymbol{A} \in \mathbb{R}^{r \times q}$:

$$\underset{\boldsymbol{B}, \boldsymbol{V} \in \mathbb{R}^{p \times k}}{\text{minimize}} \quad \|\boldsymbol{B} - \boldsymbol{A}\|_{\mathrm{F}}^2 + \eta \text{tr}\left(\boldsymbol{V}^\top \boldsymbol{L} \boldsymbol{V}\right), \text{ subject to } \boldsymbol{B} \geq \boldsymbol{0}, \ \boldsymbol{B}\boldsymbol{1}_q = \boldsymbol{1}_r, \ \boldsymbol{V}^\top \boldsymbol{V} = \boldsymbol{I}_k, \quad (3)$$

where $\boldsymbol{L}$ depends on $\boldsymbol{B}$ through (2), $\eta > 0$ is a hyperparameter that promotes the rank of $\boldsymbol{L}$ to be $p - k$, and $\boldsymbol{A}$ can be constructed from the correlation between nodes of objects and classes.

An alternating minimization approach was proposed in [12] to solve Problem (3). A shortcoming of this formulation is it lacks statistical support, which makes it difficult to draw statistical conclusions about the learned graph as well as to consider this model under different statistical assumptions.

**Spectral Regularization**. Properties associated with the spectral decomposition of graph matrices have demonstrated advantages that enable learning graphs with specific structures, such as bipartite and $k$-component graphs [11–14, 31]. The authors in [14] proposed a formulation to learn $k$-component bipartite graphs based on the GMRF framework. Their approach consists of including spectral decompositions of the Laplacian and adjacency matrices into the objective function as regularization terms. Although the formulation in [14] is derived from an statistical approach, we note that it does not always guarantee that the estimated graph is bipartite, further requiring postprocessing of the adjacency matrix. In addition, the lack of constraints on the node degrees allow trivial solutions with isolated nodes, which is also undesired in practice. The method in [14] does not require knowledge of the partition of the node set, however it does assume that the number of nodes in each set is known.

## 3 Proposed Formulations & Algorithms

In what follows, we present our proposed methods and algorithms, starting off with the simple case of a connected bipartite graph under Gaussian settings. It is important to look at this particular scenario first, as the proposed algorithm will serve as a building block for more elaborate formulations that include additional statistical assumptions, such as heavy tails, and its extension to $k$-component bipartite graphs.

### 3.1 Gaussian Bipartite Graphs

In this section, we propose a formulation to learn connected bipartite graphs from data along with an algorithm to find a global optimum. Noting that the adjacency matrix of a bipartite graph can be partitioned as $\boldsymbol{W} = [\boldsymbol{0} \ \boldsymbol{B}; \boldsymbol{B}^\top \ \boldsymbol{0}]$, where $\boldsymbol{B} \in \mathbb{R}_+^{r \times q}$, we can formulate its MLE as the following convex optimization program:

$$\underset{\boldsymbol{L}, \boldsymbol{B} \geq \boldsymbol{0}}{\text{minimize}} \quad \text{tr}\left(\boldsymbol{L}\boldsymbol{S}\right) - \log\det\left(\boldsymbol{L} + \boldsymbol{J}\right), \text{ subject to } \boldsymbol{L} = \begin{bmatrix} \text{Diag}\left(\boldsymbol{B}\boldsymbol{1}_q\right) & -\boldsymbol{B} \\ -\boldsymbol{B}^\top & \text{Diag}\left(\boldsymbol{B}^\top \boldsymbol{1}_r\right) \end{bmatrix}, \quad (4)$$

where we have used the fact that, for a connected graph, $\det^*(\boldsymbol{L}) = \det\left(\boldsymbol{L} + \boldsymbol{J}\right)$, $\boldsymbol{J} \triangleq \frac{1}{p}\boldsymbol{1}\boldsymbol{1}^\top$ [29].

We can reformulate Problem (4) by taking advantage of the symmetry of the similarity matrix $\boldsymbol{S}$, *i.e.*, $\boldsymbol{S} = [\boldsymbol{S}_{rr} \ \boldsymbol{S}_{rq}; \boldsymbol{S}_{rq}^\top \ \boldsymbol{S}_{qq}]$ where $\boldsymbol{S}_{rq} \in \mathbb{R}^{r \times q}$ contains the pairwise similarities between the nodes in the set of objects $\mathcal{V}_r$ and the nodes in the set of classes $\mathcal{V}_q$. In particular, we have

$$\text{tr}\left(\boldsymbol{L}\boldsymbol{S}\right) = \text{diag}(\boldsymbol{S})^\top \begin{bmatrix} \boldsymbol{B}\boldsymbol{1}_q \\ \boldsymbol{B}^\top \boldsymbol{1}_r \end{bmatrix} - 2\text{tr}(\boldsymbol{B}\boldsymbol{S}_{rq}^\top) = \text{tr}\left(\boldsymbol{B}\left(\boldsymbol{1}_q \boldsymbol{s}_{1:r}^\top + \boldsymbol{s}_{r+1:p}\boldsymbol{1}_r^\top - 2\boldsymbol{S}_{rq}^\top\right)\right), \quad (5)$$

where $\boldsymbol{s} \triangleq \text{diag}(\boldsymbol{S})$.

Plugging (2) and (5) in (4), we arrive at the following formulation to learn a connected bipartite graph:

$$\begin{aligned} \underset{\boldsymbol{B} \geq \boldsymbol{0}}{\text{minimize}} \quad & \text{tr}\left(\boldsymbol{B}\left(\boldsymbol{1}_q \boldsymbol{s}_{1:r}^\top + \boldsymbol{s}_{r+1:p}\boldsymbol{1}_r^\top - 2\boldsymbol{S}_{rq}^\top\right)\right) \\ & - \log\det\left(\begin{bmatrix} \text{Diag}(\boldsymbol{B}\boldsymbol{1}_q) + \boldsymbol{J}_{rr} & -\boldsymbol{B} + \boldsymbol{J}_{rq} \\ -\boldsymbol{B}^\top + \boldsymbol{J}_{qr} & \text{Diag}(\boldsymbol{B}^\top \boldsymbol{1}_r) + \boldsymbol{J}_{qq} \end{bmatrix}\right). \end{aligned} \quad (6)$$

### 3.1.1 PGD Solution

We propose an optimization algorithm based on the PGD framework [32, 33] that amounts to two steps: (1) computation of a descent update and (2) projection onto the feasible set.

A common way to compute a descent direction is using the gradient of the objective function. The gradient of the objective function in Problem (6) would involve the computation of the inverse of the matrix inside the log-determinant term, which in general costs $O(p^3)$. By leveraging the block structure of the Laplacian matrix of bipartite graphs, and using the classical matrix determinant lemma for block matrices that relates the determinant of a block matrix with its Schur complement [34, pp. 4], we have the following equality:

$$\log \det \left( \begin{bmatrix} \mathsf{Diag}(\boldsymbol{B1}_q) + \boldsymbol{J}_{rr} & -\boldsymbol{B} + \boldsymbol{J}_{rq} \\ -\boldsymbol{B}^\top + \boldsymbol{J}_{qr} & \mathsf{Diag}(\boldsymbol{B}^\top \boldsymbol{1}_r) + \boldsymbol{J}_{qq} \end{bmatrix} \right) = \log \det \left( \mathsf{Diag}(\boldsymbol{B1}_q) + \boldsymbol{J}_{rr} \right) +$$
$$\log \det \left( \mathsf{Diag}(\boldsymbol{B}^\top \boldsymbol{1}_r) + \boldsymbol{J}_{qq} - (-\boldsymbol{B}^\top + \boldsymbol{J}_{qr})(\mathsf{Diag}(\boldsymbol{B1}_q) + \boldsymbol{J}_{rr})^{-1}(-\boldsymbol{B} + \boldsymbol{J}_{rq}) \right). \quad (7)$$

In practice, we also would like to normalize the degrees of the nodes in the objects group, so that the edge weights can be directly interpreted as the degree of membership of an object to a particular class. Mathematically, this can be achieved by a simple linear constraint $\boldsymbol{B1}_q = \boldsymbol{1}_r$.

Plugging the above constraint and equality (7) into (6), we have the following formulation

$$\underset{\boldsymbol{B} \geq \boldsymbol{0}, \boldsymbol{B1}_q = \boldsymbol{1}_r}{\text{minimize}} \ \mathsf{tr}\left( \boldsymbol{BC} \right) - \log \det \left( \mathsf{Diag}(\boldsymbol{B}^\top \boldsymbol{1}_r) + \boldsymbol{J}_{qq} - (\boldsymbol{B} - \boldsymbol{J}_{rq})^\top (\boldsymbol{I}_r + \boldsymbol{J}_{rr})^{-1}(\boldsymbol{B} - \boldsymbol{J}_{rq}) \right), \quad (8)$$

where $\boldsymbol{C} \triangleq \left( \boldsymbol{s}_{r+1:p} \boldsymbol{1}_r^\top - 2\boldsymbol{S}_{rq}^\top \right)$.

Let $f(\boldsymbol{B}) \triangleq -\log \det(g(\boldsymbol{B})) + \mathsf{tr}\left( \boldsymbol{BC} \right)$ be the objective function of Problem (8), where $g(\boldsymbol{B}) \triangleq \mathsf{Diag}(\boldsymbol{B}^\top \boldsymbol{1}_r) + \boldsymbol{J}_{qq} - (\boldsymbol{B} - \boldsymbol{J}_{rq})^\top (\boldsymbol{I}_r + \boldsymbol{J}_{rr})^{-1}(\boldsymbol{B} - \boldsymbol{J}_{rq})$. Then the gradient of $f(\boldsymbol{B})$ can be computed as

$$\nabla f(\boldsymbol{B}) = \boldsymbol{1}_r \left[ \mathsf{diag}\left( -g(\boldsymbol{B})^{-1} \right) \right]^\top - 2(\boldsymbol{I}_r + \boldsymbol{J}_{rr})^{-1}(\boldsymbol{B} - \boldsymbol{J}_{rq}) \left( -g(\boldsymbol{B})^{-1} \right)^\top + \boldsymbol{C}^\top. \quad (9)$$

The PGD update is formulated as

$$\boldsymbol{B}^{l+1} = \underset{\boldsymbol{B} \geq \boldsymbol{0}, \boldsymbol{B1}_q = \boldsymbol{1}_r}{\arg \min} \ \left\| \boldsymbol{B} - \left( \boldsymbol{B}^l - \alpha_l \nabla f(\boldsymbol{B}^l) \right) \right\|_{\mathrm{F}}^2 \triangleq P_\triangle \left( \boldsymbol{B}^l - \alpha_l \nabla f(\boldsymbol{B}^l) \right). \quad (10)$$

Problem (10) is an Euclidean projection of the rows of $\boldsymbol{B}^l - \alpha_l \nabla f(\boldsymbol{B}^l)$ onto the probability simplex. The unique solution to Problem (10) can be found efficiently via several algorithms [35–37] whose theoretical worst-case complexity is $O(rq^2)$ but their observed practical complexity is $O(rq)$ [38].

Finally, to validate the point $\boldsymbol{B}^{l+1}$ and to adaptively update the learning rate $\alpha_l$, we check the following backtracking condition [33]:

$$f\left( \boldsymbol{B}^{l+1} \right) \leq f\left( \boldsymbol{B}^l \right) + \left\langle \nabla f\left( \boldsymbol{B}^l \right), \boldsymbol{B}^{l+1} - \boldsymbol{B}^l \right\rangle + \frac{1}{2\alpha_l} \| \boldsymbol{B}^{l+1} - \boldsymbol{B}^l \|_{\mathrm{F}}^2. \quad (11)$$

In practice, we decrease the learning rate $\alpha_l \in (0, 1)$ until condition (11) is satisfied or convergence has been achieved. Algorithm 1 summarizes the proposed scheme for learning a bipartite graph from a Gaussian MRF assumption. Since the objective function of Problem (8) is convex and its feasible set is compact, Algorithm 1 converges to a global minimum [32].

### 3.2 Student-*t* Bipartite Graphs

Outliers and heavy-tailed events are pervasive in modern datasets [39]. To account for this phenomena, instead of assuming a Gaussian generative process, as done in [14] and in Problem (4), we assume the data generating process to be modeled by a multivariate zero-mean Student-$t$ distribution, whose probability density function can be written as

$$p(\boldsymbol{x}) \propto \sqrt{\det^*(\boldsymbol{\Theta})} \left( 1 + \frac{\boldsymbol{x}^\top \boldsymbol{\Theta} \boldsymbol{x}}{\nu} \right)^{-(\nu+p)/2}, \quad (12)$$

---

**Algorithm 1:** Gaussian bipartite graph learning (GBG)

---

**Data:** Similarity matrix $\boldsymbol{S}$, initial feasible estimate of the graph weights $\boldsymbol{B}^0$, initial learning rate $\alpha_0 > 0$, tolerance $\epsilon > 0$.

**Result:** Optimal bipartite graph: $\boldsymbol{B}^\star$

**1 while** $l \leq$ maxiter **do**

**2** $\quad \triangleright \boldsymbol{B}^{l+1} \leftarrow P_\triangle \left( \boldsymbol{B}^l - \alpha_l \nabla f(\boldsymbol{B}^l) \right)$, where $\alpha_l$ is chosen such that (11) is satisfied

**3** $\quad$ **if** $\|\boldsymbol{B}^{l+1} - \boldsymbol{B}^l\|_\mathrm{F} / \|\boldsymbol{B}^l\|_\mathrm{F} \leq \epsilon$ **then**

**4** $\quad\quad$ **return** $\boldsymbol{B}^{l+1}$

**5** $\quad$ **end**

**6 end**

---

where $\boldsymbol{\Theta}$ is a positive-semidefinite inverse scatter matrix modeled as a combinatorial graph Laplacian matrix, and $\nu > 2$ is the number of degrees of freedom that controls the rate of decay of the tails. More precisely, as $\nu \to \infty$ the tails of (12) approach those of a Gaussian distribution.

This results in a robustified version of the MLE for connected graph learning, *i.e.*,

$$
\begin{aligned}
\underset{\boldsymbol{L}, \boldsymbol{B}}{\text{minimize}} \quad & -\log \det (\boldsymbol{L} + \boldsymbol{J}) + \frac{p+\nu}{n} \sum_{i=1}^n \log \left( 1 + \tfrac{1}{\nu} \boldsymbol{x}_i^\top \boldsymbol{L} \boldsymbol{x}_i \right), \\
\text{subject to} \quad & \boldsymbol{L} = \begin{bmatrix} \boldsymbol{I}_r & -\boldsymbol{B} \\ -\boldsymbol{B}^\top & \mathsf{Diag}\left(\boldsymbol{B}^\top \boldsymbol{1}_r\right) \end{bmatrix}, \ \boldsymbol{B} \geq \boldsymbol{0}, \ \boldsymbol{B}\boldsymbol{1}_q = \boldsymbol{1}_r.
\end{aligned}
\tag{13}
$$

We can further simplify the objective function of Problem (13) by noting that

$$
\boldsymbol{x}_i^\top \boldsymbol{L} \boldsymbol{x}_i = \boldsymbol{x}_i^\top \begin{bmatrix} \boldsymbol{I}_r & -\boldsymbol{B} \\ -\boldsymbol{B}^\top & \mathsf{Diag}\left(\boldsymbol{B}^\top \boldsymbol{1}_r\right) \end{bmatrix} \boldsymbol{x}_i = h_i + \mathsf{tr}\left(\boldsymbol{B}\boldsymbol{G}_i\right),
\tag{14}
$$

where $\boldsymbol{G}_i \triangleq \mathsf{diag}(\boldsymbol{x}_i \boldsymbol{x}_i^\top)_{r+1:p} \boldsymbol{1}_r^\top - 2 \left(\boldsymbol{x}_i \boldsymbol{x}_i^\top\right)_{rq}^\top$ and $h_i \triangleq \boldsymbol{1}_r^\top \mathsf{diag}(\boldsymbol{x}_i \boldsymbol{x}_i^\top)_{1:r}$.

Plugging (7) and (14) into (13), we have the following optimization program to learn a heavy-tailed bipartite graph:

$$
\begin{aligned}
\underset{\boldsymbol{B} \geq \boldsymbol{0}, \ \boldsymbol{B}\boldsymbol{1}_q = \boldsymbol{1}_r}{\text{minimize}} \quad & -\log \det \left( \mathsf{Diag}(\boldsymbol{B}^\top \boldsymbol{1}_r) + \boldsymbol{J}_{qq} - (\boldsymbol{B} - \boldsymbol{J}_{rq})^\top (\boldsymbol{I}_r + \boldsymbol{J}_{rr})^{-1} (\boldsymbol{B} - \boldsymbol{J}_{rq}) \right) \\
& + \frac{p+\nu}{n} \sum_{i=1}^n \log \left( 1 + \frac{h_i + \mathsf{tr}\left(\boldsymbol{B}\boldsymbol{G}_i\right)}{\nu} \right).
\end{aligned}
\tag{15}
$$

Problem (15) is, in general, nonconvex due to the summation over concave terms and hence it is difficult to be dealt with directly. To tackle this issue, we leverage the MM framework that generates a sequence of updates, where in each iteration an upper-bounded subproblem is solved [40].

To find such upper-bound, we use the fact that the logarithm term in Problem (15) can be globally upper-bounded by its first-order Taylor expansion, *i.e.*, for all $a \geq 0, t \geq 0, b > 0$, we have: $\log \left( 1 + \frac{t}{b} \right) \leq \log \left( 1 + \frac{a}{b} \right) + \frac{t-a}{a+b}$, hence at a point $\boldsymbol{B}^j$ we have that

$$
\log \left( 1 + \frac{h_i + \mathsf{tr}\left(\boldsymbol{B}\boldsymbol{G}_i\right)}{\nu} \right) \leq \frac{\mathsf{tr}\left(\boldsymbol{B}\boldsymbol{G}_i\right)}{\nu + h_i + \mathsf{tr}\left(\boldsymbol{B}^j \boldsymbol{G}_i\right)} + c,
$$

where $c \triangleq \log \left( 1 + \frac{h_i + \mathsf{tr}(\boldsymbol{B}^j \boldsymbol{G}_i)}{\nu} \right) - \frac{\mathsf{tr}(\boldsymbol{B}^j \boldsymbol{G}_i)}{\nu + h_i + \mathsf{tr}(\boldsymbol{B}^j \boldsymbol{G}_i)}$ is a constant that does not depend on $\boldsymbol{B}$.

Hence, to find an update $\boldsymbol{B}^{l+1}$, we iterate the solution of the following upper-bounded convex subproblem:

$$
\begin{aligned}
\boldsymbol{B}^{j+1} = \underset{\boldsymbol{B}}{\arg\min} \quad & -\log \det \left( \mathsf{Diag}(\boldsymbol{B}^\top \boldsymbol{1}_r) + \boldsymbol{J}_{qq} - (\boldsymbol{B} - \boldsymbol{J}_{rq})^\top (\boldsymbol{I}_r + \boldsymbol{J}_{rr})^{-1} (\boldsymbol{B} - \boldsymbol{J}_{rq}) \right) \\
& + \mathsf{tr}\left(\boldsymbol{B}\boldsymbol{M}^j\right), \text{ subject to } \boldsymbol{B} \geq \boldsymbol{0}, \ \boldsymbol{B}\boldsymbol{1}_q = \boldsymbol{1}_r,
\end{aligned}
\tag{16}
$$

where $\boldsymbol{M}^j \triangleq \frac{p+\nu}{n} \sum_{i=1}^n \frac{\boldsymbol{G}_i}{\nu + h_i + \mathrm{tr}\left(\boldsymbol{B}^j \boldsymbol{G}_i\right)}$. We observe in practice that a few ($\approx 5$) iterations are sufficient for convergence.

Problem (16) has the same format as Problem (8) and it can be readily solved by Algorithm 1. The proposed learning scheme for Student-$t$ bipartite graphs is summarized in Algorithm 2. The sequence $\left\{\boldsymbol{B}^l\right\}$ generated by Algorithm 2 converges to the set of stationary points of Problem (15). This result directly follows from the convergence results of the MM framework in [40].

---

**Algorithm 2:** Student-$t$ bipartite graph learning (SBG)

---

**Data:** Data matrix $\boldsymbol{X}$, initial feasible estimate of the graph weights $\boldsymbol{B}^0$, initial learning rate $\alpha_0 > 0$, tolerance $\epsilon > 0$, degree of freedoms $\nu > 2$.
**Result:** Learned bipartite graph: $\boldsymbol{B}^{l+1}$
1 **while** $l \leq$ maxiter **do**
2     $\triangleright$ update $\boldsymbol{B}^{l+1}$ by iterating (16)
3     **if** $\|\boldsymbol{B}^{l+1} - \boldsymbol{B}^l\|_{\mathrm{F}}/\|\boldsymbol{B}^l\|_{\mathrm{F}} \leq \epsilon$ **then**
4        **return** $\boldsymbol{B}^{l+1}$
5     **end**
6 **end**

---

### 3.3 $k$-component Student-$t$ Bipartite Graphs

Graphs with multiple components can be learned by introducing rank constraints on the Laplacian matrix [11, 12, 14]. More precisely, the number of zero eigenvalues of the Laplacian matrix amounts to the number of graph components, *i.e.*, $\mathrm{rank}(\boldsymbol{L}) = p - k$, where $k$ is the number of components.

The approach taken by [12] approximates the rank constraint by applying Fan's theorem [41] as follows

$$\sum_{i=1}^k \lambda_i\left(\boldsymbol{L}\right) = \underset{\boldsymbol{V} \in \mathbb{R}^{p \times k}, \boldsymbol{V}^\top \boldsymbol{V} = \boldsymbol{I}_k}{\mathrm{minimize}} \mathrm{tr}\left(\boldsymbol{V}^\top \boldsymbol{L} \boldsymbol{V}\right), \tag{17}$$

where the right hand side of (17) can be used as a regularization term (*cf.* Problem (3)). On the other hand, the authors in [14] introduced the quadratic relaxation $\left\|\boldsymbol{L} - \boldsymbol{U}\boldsymbol{\Lambda}\boldsymbol{U}^\top\right\|_{\mathrm{F}}^2$. Those formulations [12, 14] can be conveniently solved by coordinate descent methods. However, they introduce additional variables to be optimized, extra hyperparameters to be tuned, and do not guarantee that the rank constraint on $\boldsymbol{L}$ is satisfied.

In our proposed method, we directly apply the rank constraint into the optimization formulation. More precisely, using the Student-$t$ case, we formulate the following optimization problem to learn a $k$-component bipartite graph:

$$\begin{aligned}
\underset{\boldsymbol{L} \succeq \boldsymbol{0}, \boldsymbol{B}}{\mathrm{minimize}} \quad & \frac{p+\nu}{n} \sum_{i=1}^n \log\left(1 + \frac{h_i + \mathrm{tr}\left(\boldsymbol{B}\boldsymbol{G}_i\right)}{\nu}\right) - \log \det{}^*\left(\boldsymbol{L}\right), \\
\text{subject to} \quad & \boldsymbol{L} = \begin{bmatrix} \boldsymbol{I}_r & -\boldsymbol{B} \\ -\boldsymbol{B}^\top & \mathrm{Diag}\left(\boldsymbol{B}^\top \boldsymbol{1}_r\right) \end{bmatrix}, \ \mathrm{rank}(\boldsymbol{L}) = p - k, \ \boldsymbol{B} \geq \boldsymbol{0}, \boldsymbol{B}\boldsymbol{1}_q = \boldsymbol{1}_r.
\end{aligned} \tag{18}$$

Unlike the case for connected bipartite graphs in Problem (6), the pseudo-determinant term in (18) cannot be simplified through the block matrix determinant lemma [34]. Therefore, we design an iterative algorithm based on the ADMM framework [42] for Problem (18).

#### 3.3.1 ADMM Solution

The augmented Lagrangian of Problem (18) can be written as

$$\begin{aligned}
L_\rho(\boldsymbol{L}, \boldsymbol{B}, \boldsymbol{Y}) = & \frac{p+\nu}{n} \sum_{i=1}^n \log\left(1 + \frac{h_i + \mathrm{tr}\left(\boldsymbol{B}\boldsymbol{G}_i\right)}{\nu}\right) - \log \det{}^*\left(\boldsymbol{L}\right) \\
& + \left\langle \boldsymbol{L} - \begin{bmatrix} \boldsymbol{I}_r & -\boldsymbol{B} \\ -\boldsymbol{B}^\top & \mathrm{Diag}\left(\boldsymbol{B}^\top \boldsymbol{1}_r\right) \end{bmatrix}, \boldsymbol{Y} \right\rangle + \frac{\rho}{2} \left\| \boldsymbol{L} - \begin{bmatrix} \boldsymbol{I}_r & -\boldsymbol{B} \\ -\boldsymbol{B}^\top & \mathrm{Diag}\left(\boldsymbol{B}^\top \boldsymbol{1}_r\right) \end{bmatrix} \right\|_{\mathrm{F}}^2,
\end{aligned} \tag{19}$$

where $\rho > 0$ is a penalty hyperparameter.

For fixed $\boldsymbol{B}$ and $\boldsymbol{Y}$, the subproblem for $\boldsymbol{L}$ can be written as

$$\boldsymbol{L}^\star = \underset{\mathrm{rank}(\boldsymbol{L})=p-k}{\arg\min} \; -\log\det{}^*(\boldsymbol{L}) + \langle \boldsymbol{L}, \boldsymbol{Y} \rangle + \frac{\rho}{2} \left\| \boldsymbol{L} - \begin{bmatrix} \boldsymbol{I}_r & -\boldsymbol{B} \\ -\boldsymbol{B}^\top & \mathrm{Diag}\left(\boldsymbol{B}^\top \boldsymbol{1}_r\right) \end{bmatrix} \right\|_{\mathrm{F}}^2, \tag{20}$$

whose closed-form solution is given by Lemma 1.

**Lemma 1** *The global minimizer of problem* (20) *is [43, 44]*

$$\boldsymbol{L}^\star = \frac{1}{2\rho} \boldsymbol{R} \left( \boldsymbol{\Gamma} + \sqrt{\boldsymbol{\Gamma}^2 + 4\rho \boldsymbol{I}} \right) \boldsymbol{R}^\top, \tag{21}$$

*where $\boldsymbol{R}\boldsymbol{\Gamma}\boldsymbol{R}^\top$ is the eigenvalue decomposition of $\rho \begin{bmatrix} \boldsymbol{I}_r & -\boldsymbol{B} \\ -\boldsymbol{B}^\top & \mathrm{Diag}\left(\boldsymbol{B}^\top \boldsymbol{1}_r\right) \end{bmatrix} - \boldsymbol{Y}$, with $\boldsymbol{\Gamma}$ having the largest $p-k$ eigenvalues along its diagonal and $\boldsymbol{R} \in \mathbb{R}^{p \times (p-k)}$ containing the corresponding eigenvectors.*

Fixing $\boldsymbol{L}$ and $\boldsymbol{Y}$, the subproblem for $\boldsymbol{B}$ is

$$\begin{aligned}
\boldsymbol{B}^\star = \underset{\boldsymbol{B} \geq \boldsymbol{0}, \boldsymbol{B}\boldsymbol{1}=\boldsymbol{1}}{\arg\min} \; & \frac{p+\nu}{n} \sum_{i=1}^n \log\left(1 + \frac{h_i + \mathrm{tr}\left(\boldsymbol{B}\boldsymbol{G}_i\right)}{\nu}\right) + \frac{\rho}{2} \left\| \boldsymbol{L} - \begin{bmatrix} \boldsymbol{I}_r & -\boldsymbol{B} \\ -\boldsymbol{B}^\top & \mathrm{Diag}\left(\boldsymbol{B}^\top \boldsymbol{1}_r\right) \end{bmatrix} \right\|_{\mathrm{F}}^2 \\
& - \left\langle \begin{bmatrix} \boldsymbol{I}_r & -\boldsymbol{B} \\ -\boldsymbol{B}^\top & \mathrm{Diag}\left(\boldsymbol{B}^\top \boldsymbol{1}_r\right) \end{bmatrix}, \boldsymbol{Y} \right\rangle,
\end{aligned} \tag{22}$$

which can be simplified as

$$\boldsymbol{B}^\star = \underset{\boldsymbol{B} \geq \boldsymbol{0}, \boldsymbol{B}\boldsymbol{1}=\boldsymbol{1}}{\arg\min} \; \frac{p+\nu}{n} \sum_{i=1}^n \log\left(1 + \frac{h_i + \mathrm{tr}\left(\boldsymbol{B}\boldsymbol{G}_i\right)}{\nu}\right) + \mathrm{tr}\left(\boldsymbol{B}\boldsymbol{H}\right) + \rho\|\boldsymbol{B}\|_{\mathrm{F}}^2 + \frac{\rho}{2}\boldsymbol{1}_r^\top \boldsymbol{B}\boldsymbol{B}^\top \boldsymbol{1}_r \tag{23}$$

where $\boldsymbol{H} \triangleq 2\boldsymbol{Y}_{rq}^\top - \boldsymbol{y}_{r+1:p}\boldsymbol{1}_r^\top - \rho\left(\boldsymbol{l}_{r+1:p}\boldsymbol{1}_r^\top - 2\boldsymbol{L}_{rq}^\top\right)$, $\boldsymbol{y}_{r+1:p} \triangleq \mathrm{diag}\left(\boldsymbol{Y}\right)_{r+1:p}$, and $\boldsymbol{l}_{r+1:p} \triangleq \mathrm{diag}\left(\boldsymbol{L}\right)_{r+1:p}$.

Leveraging the MM framework to Problem (23), like it was applied to Problem (15), we can find an upperbounded subproblem at point $\boldsymbol{B}^j$ as follows:

$$\boldsymbol{B}^{j+1} = \underset{\boldsymbol{B} \geq \boldsymbol{0}, \boldsymbol{B}\boldsymbol{1}=\boldsymbol{1}}{\arg\min} \; \mathrm{tr}\left(\boldsymbol{B}\left(\boldsymbol{H} + \boldsymbol{M}^j\right)\right) + \rho\|\boldsymbol{B}\|_{\mathrm{F}}^2 + \frac{\rho}{2}\boldsymbol{1}_r^\top \boldsymbol{B}\boldsymbol{B}^\top \boldsymbol{1}_r. \tag{24}$$

Subproblem (24) is strongly convex, thus it can be solved efficiently via convex solvers. Since the objective function of (24) is smooth and its feasible set is compact, we provide a PGD algorithm in the Supplementary Material. Finally, to find an update $\boldsymbol{B}^\star$, we iterate the solution of Problem (24). The dual variable $\boldsymbol{Y}$ is updated approximately via gradient ascent.

Algorithm 3 summarizes our proposed scheme to learn $k$-component bipartite graphs and its convergence is established through Theorem 2, whose proof is presented in the Supplementary Material.

**Theorem 2** *Algorithm 3 converges subsequently for any sufficiently large $\rho$, that is, the sequence $\left\{\left(\boldsymbol{L}^l, \boldsymbol{B}^l, \boldsymbol{Y}^l\right)\right\}$ generated by Algorithm 3 has at least one limit point, and each limit point is a stationary point of* (18).

## 4   Experimental Results

We conduct experiments to illustrate the performance of the proposed algorithms in terms of quantitative measures such as node label accuracy and graph modularity. Accuracy is computed as the ratio between the number of correctly predicted node labels and the number of nodes in the objects set, whereas modularity measures the strength of division of a graph into groups [45]. A high modularity value means that the nodes from the same group are more likely to be connected.

---

**Algorithm 3:** Student-$t$ $k$-component bipartite graph learning (kSBG)

---

**Data:** Data matrix $\boldsymbol{X}$, number of components $k$, initial feasible estimate of the graph weights $\boldsymbol{B}^0$, penalty hyperparameter $\rho > 0$, tolerance $\epsilon > 0$, degrees of freedom $\nu > 2$.

**Result:** Learned bipartite graph: $\boldsymbol{B}^{l+1}$

1 **while** $l \leq$ maxiter **do**
2     $\triangleright$ update $\boldsymbol{L}^{l+1}$ via (21)
3     $\triangleright$ update $\boldsymbol{B}^{l+1}$ by iterating (24)
4     $\triangleright$ update $\boldsymbol{Y}^{l+1} = \boldsymbol{Y}^l - \rho \left( \boldsymbol{L}^{l+1} - \begin{bmatrix} \boldsymbol{I}_r & -\boldsymbol{B}^{l+1} \\ -\boldsymbol{B}^{l+1\top} & \mathrm{Diag}\left( \boldsymbol{B}^{l+1\top} \mathbf{1}_r \right) \end{bmatrix} \right)$
5     **if** $\|\boldsymbol{B}^{l+1} - \boldsymbol{B}^l\|_{\mathrm{F}} / \|\boldsymbol{B}^l\|_{\mathrm{F}} \leq \epsilon$ **then**
6        **return** $\boldsymbol{B}^{l+1}$
7     **end**
8 **end**

---

*Benchmarks*: We compare the proposed algorithms with state-of-the-art methods in two settings: (*i*) connected bipartite graphs, where SOBG ($k = 1$) [12] and SGA [14] are the benchmarks, and (*ii*) $k$-component bipartite graphs, where SOBG ($k > 1$) and SGLA [14] act as benchmarks. In our Algorithm 3, we set the hyperparameter $\rho = 1$ and the relative tolerance $\epsilon = 10^{-5}$. The hyperparameter of SOBG was tunned according to the heuristic procedure described in [12]. The proposed algorithms and benchmarks were implemented using the R programming language. For SGA and SGLA, we relied on their official implementation via the package spectralGraphTopology [46].

## 4.1 Returns of S&P500 Stocks

We perform experiments using publicly available, historical daily prices, queried from Yahoo! Finance$^{\mathrm{TM}}$, of $r = 333$ stocks listed in the S&P500 Index and $q = 8$ sectors indexes, namely: Consumer Staples, Energy, Financials, Health Care, Industrials, Materials, Real Estate, and Utilities. Following our notation, the node set of classes $\mathcal{V}_q$ represents the sectors indexes, whereas the stocks are represented by the node set of objects $\mathcal{V}_r$.

Stock sector labels are available through the Global Industry Classification Standard (GICS) [47, 48]. Arguably the most well-recognized, industry-standard stock classification system, GICS relies on a fixed classification structure that may not fully capture the actual impact of a particular stock at particular time. With the intent of solving this shortcoming, we employ the learned bipartite graphs to extend current industry stock classification system from a static classification to a dynamic soft clustering, in which a particular node (stock) may belong to different clusters (sectors) with different degrees of membership that can change over time. Our goal is to perform soft clustering of stocks according to the market sectors. To that end, we recall that each row of $\boldsymbol{B}$ is a point belonging to the probability simplex in $\mathbb{R}^q$, which represents the level of membership of a stock to a particular set of sectors. The final sector assigned to the $i$-th stock corresponds to $\arg\max_{j \in 1, \ldots, q} B_{ij}$.

We start by constructing the log-returns data matrix, *i.e.*, a matrix $\boldsymbol{X} \in \mathbb{R}^{n \times p}$, where $n$ is the number of log-returns observations and $p$ is the number of instruments, as $X_{i,j} = \log P_{i,j} - \log P_{i-1,j}$, where $P_{i,j}$ is the closing price of the $j$-th instrument at the $i$-th day. We collect data from Jan 5th 2016 to Jan 5th 2021, totalling $n = 1291$ daily observations. Then, we proceed to learn connected bipartite graphs via the benchmarks and proposed methods using the observations $\boldsymbol{X}$ in a rolling-window fashion[1]. More precisely, we use a window of length 504 days (2 years in terms of stock market days) and shift this window by 63 days (3 months in terms of stock market days).

Figure 1 shows quantitative measurements that reveal the higher performance of the proposed SBG algorithm both in terms of accuracy and modularity. That may be explained by the fact that SBG adopts a multivariate Student-$t$ distribution, which is well-known to better model financial datasets in comparison to Gaussian assumptions.

---

[1]An estimate for the degree of freedoms $\nu$, required by Algorithms 2 and 3, is obtained by fitting a univariate Student-$t$ distribution to the log-returns of the S&P500 index during the corresponding time period.

Furthermore, we evaluate the performance of the $k$-component bipartite estimators under the same settings as previously described, where we set $k = 8$, *i.e.*, $k$ is the number of sectors. Figure 2 reveals that kSBG presents a higher performance accross the whole time period both in terms of accuracy and modularity when compared to SGLA and SOBG. Interestingly, in both Figure 1 and 2, we observe a decline in accuracy and modularity of the methods GBG, SBG, and kSBG, starting in Jan 2020, which can be explained by the impact of the COVID-19 pandemic on the US stock market. This reveals that our methods may be able to identify periods of economic turmoil and suggests that, during such times, standards such as GICS may not be the best reference for stock market sector classification. This insight may be critical for investors who rely on diversification of their portfolios through information on the sectors [49].

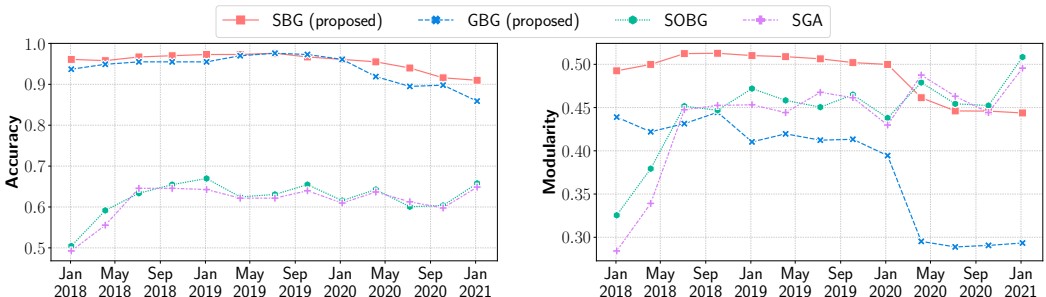

Figure 1: Performance of the estimators for connected bipartite graphs of S&P500 stocks.

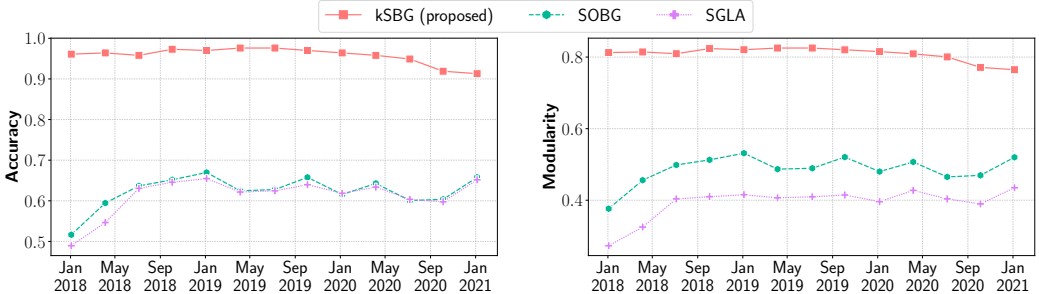

Figure 2: Performance of the estimators for 8-component bipartite graphs of S&P500 stocks.

Finally, Figure 3 shows the $k$-component bipartite models learned using the whole dataset. We observe that SGLA allows the existence of isolated nodes, which degrades performance. Our proposed method kSBG avoids that issue by imposing a linear constraint on the node degrees. In addition, kSBG yields a more intuitive representation of the financial network than SOBG as verified by the accuracy and modularity values.

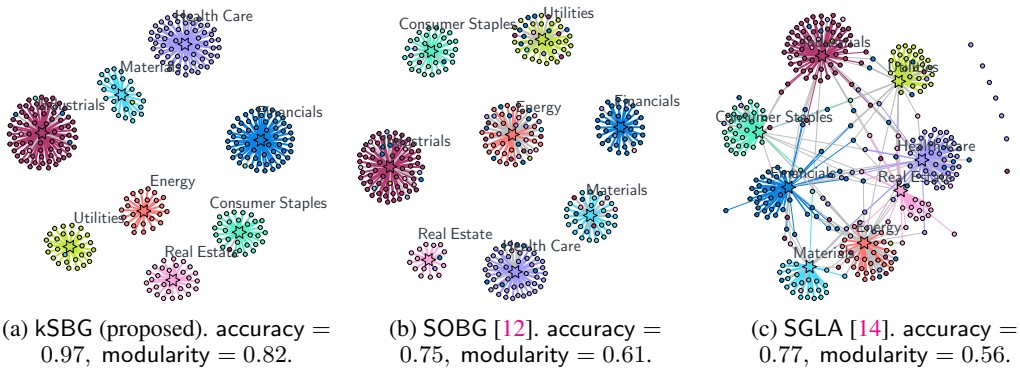

(a) kSBG (proposed). accuracy = 0.97, modularity = 0.82.

(b) SOBG [12]. accuracy = 0.75, modularity = 0.61.

(c) SGLA [14]. accuracy = 0.77, modularity = 0.56.

Figure 3: Graphs learned by different algorithms in the S&P500 stock returns dataset.

## 4.2 Robustness to Initialization

The performance of iterative algorithms may be affected by their initial point, especially when dealing with nonconvex problems as in Algorithms 2 and 3. Hence, we perform an experiment to compare two initialization schemes: (*i*) uniform, the graph weights are randomly sampled, *i.e.*, $B_{ij}^0 \sim \mathsf{Uniform}(0,1)$, and (*ii*) default, $\boldsymbol{B}^0 \propto \max(\boldsymbol{0}, -\boldsymbol{\Xi}_{rq})$, where $\boldsymbol{S}^{-1} = \left[\boldsymbol{\Xi}_{rr}\ \boldsymbol{\Xi}_{rq}; \boldsymbol{\Xi}_{rq}^\top\ \boldsymbol{\Xi}_{qq}\right]$ is the inverse of the sample correlation matrix, as discussed in more details in [14]. In both schemes, we normalize the rows of $\boldsymbol{B}^0$ such that $\boldsymbol{B}^0\boldsymbol{1} = \boldsymbol{1}$ holds. For this experiment, we consider log-returns of $r = 362$ stocks and $q = 9$ stock sectors from Oct. 5th 2005 to Dec. 30th 2015, totalling $n = 2577$ observations. Figure 4 shows the result of 100 realizations of Algorithms 2 and 3. It can be observed that Algorithms 2 and 3 are not sensitive to the choice of an initial point, which is an important feature for practical applications. Finally, Figures 5a and 5b show the learned graphs from kSBG with different initial points, where we observe that the performance of kSBG is largely unaffected.

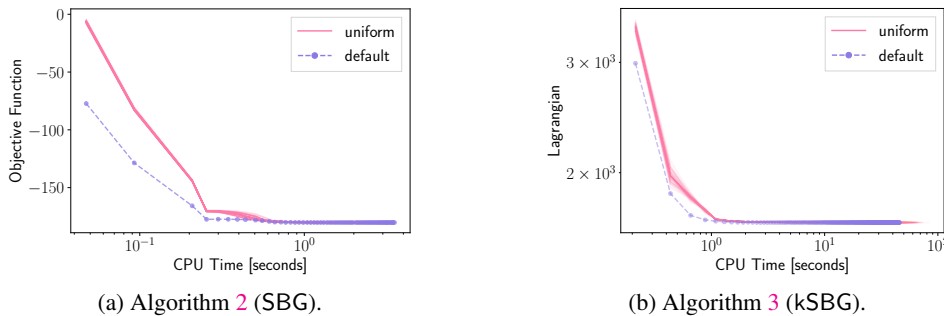

(a) Algorithm 2 (SBG).          (b) Algorithm 3 (kSBG).

Figure 4: Convergence trend of the proposed algorithms for different initial points.

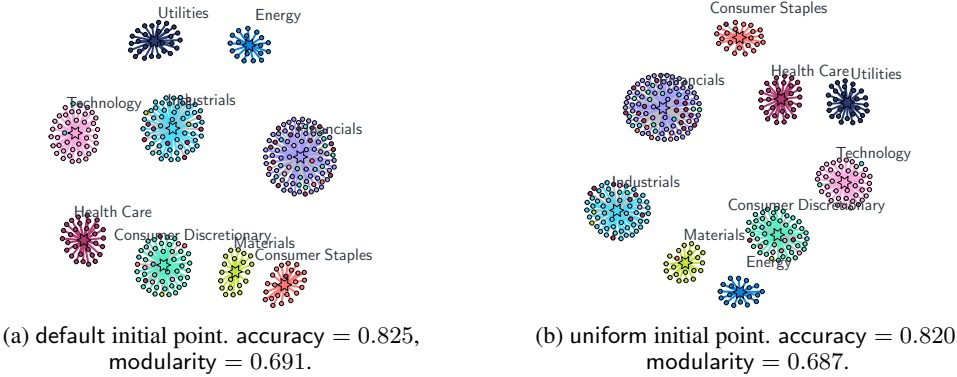

(a) default initial point. accuracy $= 0.825$,          (b) uniform initial point. accuracy $= 0.820$,
modularity $= 0.691$.                                    modularity $= 0.687$.

Figure 5: Visualizations of the graphs learned by kSBG with distinct initial points.

## 5 Conclusions

The design of numerical algorithms plays a critical role in disseminating the application of graphical models in practical, real-world instances. This paper has proposed an optimization formulation to the problem of learning a bipartite graph from data along with an algorithm based on the PGD framework. Envisioning applications in scenarios where outliers or heavy-tail events are present, we have extended the proposed formulation to the case where the data follow a multivariate Student-$t$ distribution. In addition, we have proposed an optimization formulation, along with a provably convergent ADMM-based algorithm, for estimating $k$-component bipartite graphs. Experimental results with datasets of log-returns of S&P500 stocks have provided substantial evidence for the improved performance of the proposed algorithms in comparison to state-of-the-art methods.

## Acknowledgments

This work was supported by the Hong Kong GRF 16207820 research grant.

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
