## Supplementary Material

In this supplementary document we include additional results and analysis that showcase the robustness of our proposed algorithms. More precisely, Section A contains definitions that were not formally stated in the main manuscript, Section B contains additional experimental results that reinforce the outperformance of the proposed algorithms relative to their benchmarks, Section C contains empirical convergence results of the proposed algorithms, Section D describes a detailed convergence analysis for Algorithm 3 in the form of a proof of Theorem 2 presented in the main manuscript, and Section E describes a simple projected gradient descent algorithm for Subproblem (24) stated in the main manuscript.

## A  Definitions

**Definition 3** *(Modularity) The modularity of a graph $\mathcal{G}$ [45] is defined as $Q : \mathcal{G} \to \mathbb{R}$:*

$$Q(\mathcal{G}) \triangleq \frac{1}{2|\mathcal{E}|} \sum_{i,j \in \mathcal{V}} \left( W_{ij} - \frac{d_i d_j}{2|\mathcal{E}|} \right) \mathbb{1}(t_i = t_j), \tag{25}$$

*where $d_i$ is the weighted degree of the $i$-th node, $t_i$ is the type (or label) of the $i$-th node, and $\mathbb{1}(\cdot)$ is the indicator function.*

**Definition 4** *(Relative Error) The relative error between a ground truth matrix $\mathbf{B}_{\text{true}}$ of edge weights and its estimated version $\mathbf{B}^\star$ is defined as:*

$$\text{RE}(\mathbf{B}_{\text{true}}, \mathbf{B}^\star) = \frac{\|\mathbf{B}_{\text{true}} - \mathbf{B}^\star\|_{\mathsf{F}}}{\|\mathbf{B}_{\text{true}}\|_{\mathsf{F}}}. \tag{26}$$

## B  Additional Experiments

### B.1  S&P500 Stocks

We perform additional experiments considering log-returns of $r = 362$ stocks and $q = 9$ stock sectors from Oct. 5th 2005 to Dec. 30th 2015, totalling $n = 2577$ observations. Figure 6 shows accuracy and modularity measurements of the graphs learned in a rolling window basis. More precisely, we chose a window of length 504 days (2 years in terms of stock market days) and we shift this window by 63 days (3 months in terms of stock market days). Likewise in the experiment described in the main manuscript, we observe a sharp decline in accuracy and modularity for the proposed kSBG around October 2008, which can be explained by the effect of the housing bubble crisis. Thus, our proposed method may be able to capture events in financial networks, which can further be used to reduce risks in financial tasks such as portfolio design.

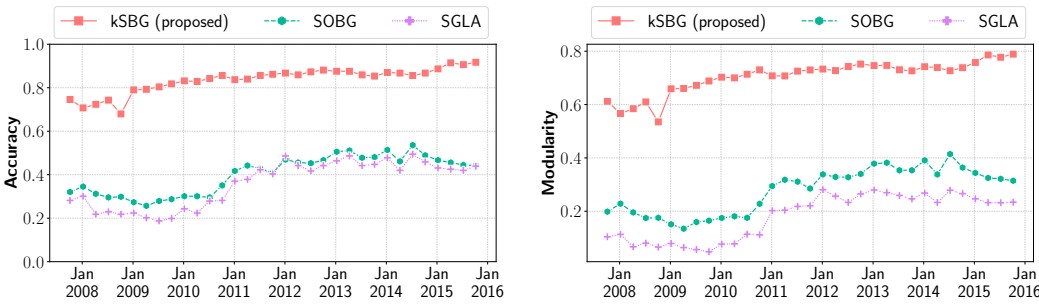

Figure 6: Measurements of accuracy and modularity between kSBG (Algorithm 3, proposed) and the competing methods SGLA [14] and SOBG [12].

### B.2  Synthetic Data

We perform an experiment with synthetic data in order to study the estimation error of the graph learning algorithms as a function of the number of samples. To that end, we generate synthetic data

from a multivariate Gaussian distribution, where the covariance matrix is set to be the pseudo-inverse of the Laplacian matrix of a bipartite graph, *i.e.*, $\boldsymbol{x} \sim \mathcal{N}\left(\boldsymbol{0}, \boldsymbol{L}_{\text{true}}^{\dagger}\right)$. Note that this setting is the same as the one described in [14]. We set the number of nodes to be $p = 110$, where $r = 100$ and $q = 10$. We then sample $n$ observations from $\boldsymbol{x}$, where we seek to measure the relative error (26) between the ground truth Laplacian matrix, $\boldsymbol{L}_{\text{true}}$, and its estimated version by different graph learning methods. Figure 7 shows the average relative error for each sample size ratio $(n/p)$ averaged over 100 realizations, where we can observe that GBG outperforms the competing methods for small and medium sample size regimes. For large sample size $(n/p \geq 50)$, the methods present a statistically similar performance.

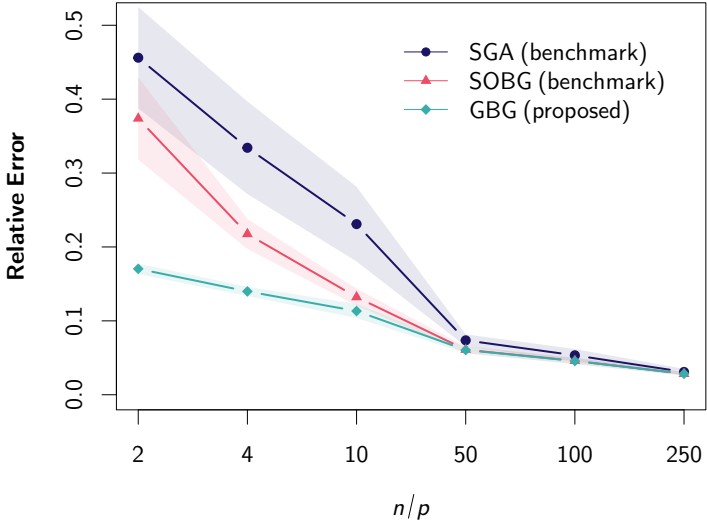

Figure 7: Relative error versus sample size ratio for bipartite graph learning algorithms under Gaussian settings. The shaded area around the solid curves represent the 1-sigma confidence interval around the average.

## C  Empirical Convergence

In this section, we illustrate the empirical convergence performance of the proposed algorithms. More precisely, we compute the augmented Lagrangian of Algorithm 3 and the objective functions of Algorithms 1 and 2 while learning a graph with the whole data matrix $\boldsymbol{X}$ that contains $p = 341$ time series with $n = 1291$ observations as described in the main manuscript. All the experiments were carried out in a MacBook Pro 13in. 2019 with Intel Core i7 2.8GHz, 16GB of RAM.

Figures 8a, 8b, and 8c show the objective functions of Problems (8) and (15), and the Lagrangian (19), respectively. In all cases, we observed a sharp improvement of the objective quantities in the first few iterations, which suggests that the algorithms create efficient updates.

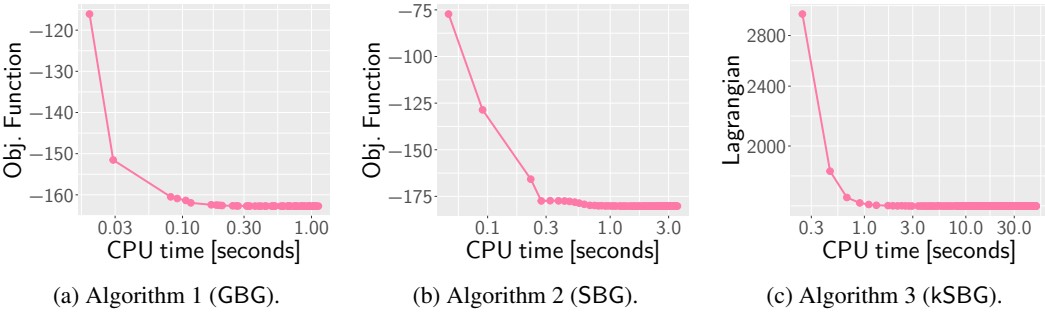

(a) Algorithm 1 (GBG).  (b) Algorithm 2 (SBG).  (c) Algorithm 3 (kSBG).

Figure 8: Empirical convergence of the proposed algorithms.

# D  Proof of Theorem 2

Since Problem (18) contains nonconvex terms in both its objective function and constraints, the convergence of Algorithm 3 is carefully analyzed and established through Theorem 2 in the main manuscript. In the proof of Theorem 2, presented herein, we borrow ideas from recent convergence results for the ADMM framework [50].

**Proof**  To prove Theorem 2, we first establish the boundedness of the sequence $\left\{\left(\boldsymbol{L}^l, \boldsymbol{B}^l, \boldsymbol{Y}^l\right)\right\}$ generated by Algorithm 3 in Lemma 5, and the monotonicity of $L_\rho\left(\boldsymbol{L}^l, \boldsymbol{B}^l, \boldsymbol{Y}^l\right)$ in Lemma 6.

**Lemma 5**  *The sequence $\left\{\left(\boldsymbol{L}^l, \boldsymbol{B}^l, \boldsymbol{Y}^l\right)\right\}$ generated by Algorithm 3 is bounded.*

**Proof**  Let $\boldsymbol{B}^0$ and $\boldsymbol{Y}^0$ be the initial points of the sequences $\left\{\boldsymbol{B}^l\right\}$ and $\left\{\boldsymbol{Y}^l\right\}$, respectively, and $\left\|\boldsymbol{B}^0\right\|_{\mathrm{F}}$ and $\left\|\boldsymbol{Y}^0\right\|_{\mathrm{F}}$ are bounded. We prove the boundedness of the sequence by induction.

Recall that the sequence $\left\{\boldsymbol{L}^l\right\}$ is established by

$$\boldsymbol{L}^l = \frac{1}{2\rho}\boldsymbol{R}^{l-1}\left(\boldsymbol{\Gamma}^{l-1} + \sqrt{\left(\boldsymbol{\Gamma}^{l-1}\right)^2 + 4\rho\boldsymbol{I}}\right)\left(\boldsymbol{R}^{l-1}\right)^\top, \qquad (27)$$

where $\boldsymbol{\Gamma}^{l-1}$ contains the largest $p-k$ eigenvalues of $\rho\begin{bmatrix}\boldsymbol{I}_r & -\boldsymbol{B}^{l-1} \\ -\left(\boldsymbol{B}^{l-1}\right)^\top & \mathsf{Diag}\left(\left(\boldsymbol{B}^{l-1}\right)^\top\boldsymbol{1}_r\right)\end{bmatrix} - \boldsymbol{Y}^{l-1}$,

and $\boldsymbol{R}^{l-1}$ contains the corresponding eigenvectors. When $l = 1$, $\left\|\boldsymbol{\Gamma}^0\right\|_{\mathrm{F}}$ is bounded since both $\left\|\boldsymbol{B}^{l-1}\right\|_{\mathrm{F}}$ and $\left\|\boldsymbol{Y}^{l-1}\right\|_{\mathrm{F}}$ are bounded. Therefore, we can conclude that $\left\|\boldsymbol{L}^1\right\|_{\mathrm{F}}$ is bounded.

Recall that the variable $\boldsymbol{B}$ satisfies the constraints $\boldsymbol{B} \geq \boldsymbol{0}$ and $\boldsymbol{B}\boldsymbol{1} = \boldsymbol{1}$, thus $\boldsymbol{B}$ is in a compact set. Therefore, $\left\|\boldsymbol{B}^l\right\|_{\mathrm{F}}$ and $\left\|\boldsymbol{V}^l\right\|_{\mathrm{F}}$ are bounded for any $l \geq 1$.

According to the dual variable update, we have

$$\boldsymbol{Y}^1 = \boldsymbol{Y}^0 - \rho\left(\boldsymbol{L}^1 - \begin{bmatrix}\boldsymbol{I}_r & -\boldsymbol{B}^1 \\ -\left(\boldsymbol{B}^1\right)^\top & \mathsf{Diag}\left(\left(\boldsymbol{B}^1\right)^\top\boldsymbol{1}_r\right)\end{bmatrix}\right). \qquad (28)$$

We can see that $\left\|\boldsymbol{Y}^1\right\|_{\mathrm{F}}$ is bounded because of the boundedness of $\left\|\boldsymbol{L}^1\right\|_{\mathrm{F}}$ and $\left\|\boldsymbol{B}^1\right\|_{\mathrm{F}}$. Therefore, it holds for $l = 1$ that $\left\{\left(\boldsymbol{L}^l, \boldsymbol{B}^l, \boldsymbol{Y}^l\right)\right\}$ is bounded.

Now we assume that $\left\{\left(\boldsymbol{L}^{l-1}, \boldsymbol{B}^{l-1}, \boldsymbol{Y}^{l-1}\right)\right\}$ is bounded for some $l \geq 1$, and check the boundedness of $\left\{\left(\boldsymbol{L}^l, \boldsymbol{B}^l, \boldsymbol{Y}^l\right)\right\}$. Similarly to the proof in (27), we can obtain that $\left\|\boldsymbol{L}^l\right\|_{\mathrm{F}}$ is bounded. We can also obtain that $\left\|\boldsymbol{Y}^l\right\|_{\mathrm{F}}$ is bounded according to the boundedness of $\left\|\boldsymbol{L}^l\right\|_{\mathrm{F}}$, $\left\|\boldsymbol{B}^l\right\|_{\mathrm{F}}$ and $\left\|\boldsymbol{Y}^{l-1}\right\|_{\mathrm{F}}$. As a result, $\left\{\left(\boldsymbol{L}^l, \boldsymbol{B}^l, \boldsymbol{Y}^l\right)\right\}$ is bounded, completing the induction. Therefore, we establish the boundedness of the sequence $\left\{\left(\boldsymbol{L}^l, \boldsymbol{B}^l, \boldsymbol{Y}^l\right)\right\}$. ∎

**Lemma 6**  *The sequence $L_\rho\left(\boldsymbol{L}^l, \boldsymbol{B}^l, \boldsymbol{Y}^l\right)$ generated by Algorithm 3 is lower bounded, and*

$$L_\rho\left(\boldsymbol{L}^{l+1}, \boldsymbol{B}^{l+1}, \boldsymbol{Y}^{l+1}\right) \leq L_\rho\left(\boldsymbol{L}^l, \boldsymbol{B}^l, \boldsymbol{Y}^l\right), \quad \forall l \in \mathbb{N}_+, \qquad (29)$$

*holds for any sufficiently large $\rho$.*

**Proof**  From expression (19) in the manuscript, we have that the augmented Lagrangian is written as

$$L_\rho(\boldsymbol{L}^l, \boldsymbol{B}^l, \boldsymbol{Y}^l) = \frac{p+\nu}{n}\sum_{i=1}^n \log\left(1 + \frac{h_i + \mathrm{tr}\left(\boldsymbol{B}^l\boldsymbol{G}_i\right)}{\nu}\right) - \log\det^*\left(\boldsymbol{L}^l\right)$$

$$+ \left\langle \boldsymbol{L}^l - \begin{bmatrix}\boldsymbol{I}_r & -\boldsymbol{B}^l \\ -\left(\boldsymbol{B}^l\right)^\top & \mathsf{Diag}\left(\left(\boldsymbol{B}^l\right)^\top\boldsymbol{1}_r\right)\end{bmatrix}, \boldsymbol{Y}^l \right\rangle + \frac{\rho}{2}\left\|\boldsymbol{L}^l - \begin{bmatrix}\boldsymbol{I}_r & -\boldsymbol{B}^l \\ -\left(\boldsymbol{B}^l\right)^\top & \mathsf{Diag}\left(\left(\boldsymbol{B}^l\right)^\top\boldsymbol{1}_r\right)\end{bmatrix}\right\|_{\mathrm{F}}^2.$$

$$(30)$$

We can see that the lower boundedness of the sequence $L_\rho\left(\boldsymbol{L}^l, \boldsymbol{B}^l, \boldsymbol{Y}^l\right)$ can be established by the boundedness of $\left\{\left(\boldsymbol{L}^l, \boldsymbol{B}^l, \boldsymbol{Y}^l\right)\right\}$ in Lemma 5.

We first establish that

$$L_\rho\left(\boldsymbol{L}^{l+1}, \boldsymbol{B}^l, \boldsymbol{Y}^l\right) \leq L_\rho\left(\boldsymbol{L}^l, \boldsymbol{B}^l, \boldsymbol{Y}^l\right), \quad \forall l \in \mathbb{N}_+. \tag{31}$$

We have

$$L_\rho(\boldsymbol{L}^{l+1}, \boldsymbol{B}^l, \boldsymbol{Y}^l) = \frac{p+\nu}{n}\sum_{i=1}^{n}\log\left(1 + \frac{h_i + \operatorname{tr}\left(\boldsymbol{B}^l\boldsymbol{G}_i\right)}{\nu}\right) - \log\det{}^*\left(\boldsymbol{L}^{l+1}\right)$$

$$+ \left\langle \boldsymbol{L}^{l+1} - \begin{bmatrix} \boldsymbol{I}_r & -\boldsymbol{B}^l \\ -\left(\boldsymbol{B}^l\right)^\top & \operatorname{Diag}\left(\left(\boldsymbol{B}^l\right)^\top\mathbf{1}_r\right) \end{bmatrix}, \boldsymbol{Y}^l \right\rangle + \frac{\rho}{2}\left\| \boldsymbol{L}^{l+1} - \begin{bmatrix} \boldsymbol{I}_r & -\boldsymbol{B}^l \\ -\left(\boldsymbol{B}^l\right)^\top & \operatorname{Diag}\left(\left(\boldsymbol{B}^l\right)^\top\mathbf{1}_r\right) \end{bmatrix} \right\|_{\mathrm{F}}^2.$$

Then we obtain

$$L_\rho(\boldsymbol{L}^{l+1}, \boldsymbol{B}^l, \boldsymbol{Y}^l) - L_\rho(\boldsymbol{L}^l, \boldsymbol{B}^l, \boldsymbol{Y}^l)$$

$$= -\log\det{}^*\left(\boldsymbol{L}^{l+1}\right) + \left\langle \boldsymbol{L}^{l+1}, \boldsymbol{Y}^l \right\rangle + \frac{\rho}{2}\left\| \boldsymbol{L}^{l+1} - \begin{bmatrix} \boldsymbol{I}_r & -\boldsymbol{B}^l \\ -\left(\boldsymbol{B}^l\right)^\top & \operatorname{Diag}\left(\left(\boldsymbol{B}^l\right)^\top\mathbf{1}_r\right) \end{bmatrix} \right\|_{\mathrm{F}}^2$$

$$- \left( -\log\det{}^*\left(\boldsymbol{L}^l\right) + \left\langle \boldsymbol{L}^l, \boldsymbol{Y}^l \right\rangle + \frac{\rho}{2}\left\| \boldsymbol{L}^l - \begin{bmatrix} \boldsymbol{I}_r & -\boldsymbol{B}^l \\ -\left(\boldsymbol{B}^l\right)^\top & \operatorname{Diag}\left(\left(\boldsymbol{B}^l\right)^\top\mathbf{1}_r\right) \end{bmatrix} \right\|_{\mathrm{F}}^2 \right).$$

Note that $\boldsymbol{L}^{l+1}$ minimizes the objective function

$$\boldsymbol{L}^{l+1} = \operatorname*{arg\,min}_{\substack{\operatorname{rank}(\boldsymbol{L})=p-k \\ \boldsymbol{L}\succeq\mathbf{0}}} -\log\det{}^*(\boldsymbol{L}) + \left\langle \boldsymbol{L}, \boldsymbol{Y}^l \right\rangle + \frac{\rho}{2}\left\| \boldsymbol{L} - \begin{bmatrix} \boldsymbol{I}_r & -\boldsymbol{B}^l \\ -\left(\boldsymbol{B}^l\right)^\top & \operatorname{Diag}\left(\left(\boldsymbol{B}^l\right)^\top\mathbf{1}_r\right) \end{bmatrix} \right\|_{\mathrm{F}}^2. \tag{32}$$

Therefore

$$L_\rho(\boldsymbol{L}^{l+1}, \boldsymbol{B}^l, \boldsymbol{Y}^l) - L_\rho(\boldsymbol{L}^l, \boldsymbol{B}^l, \boldsymbol{Y}^l) \leq 0 \tag{33}$$

holds for any $l \in \mathbb{N}_+$.

One has

$$L_\rho(\boldsymbol{L}^{l+1}, \boldsymbol{B}^l, \boldsymbol{Y}^l) - L_\rho(\boldsymbol{L}^{l+1}, \boldsymbol{B}^{l+1}, \boldsymbol{Y}^{l+1})$$

$$= \underbrace{\left\langle \boldsymbol{L}^{l+1} - \begin{bmatrix} \boldsymbol{I}_r & -\boldsymbol{B}^l \\ -\left(\boldsymbol{B}^l\right)^\top & \operatorname{Diag}\left(\left(\boldsymbol{B}^l\right)^\top\mathbf{1}_r\right) \end{bmatrix}, \boldsymbol{Y}^l \right\rangle - \left\langle \boldsymbol{L}^{l+1} - \begin{bmatrix} \boldsymbol{I}_r & -\boldsymbol{B}^{l+1} \\ -\left(\boldsymbol{B}^{l+1}\right)^\top & \operatorname{Diag}\left(\left(\boldsymbol{B}^{l+1}\right)^\top\mathbf{1}_r\right) \end{bmatrix}, \boldsymbol{Y}^{l+1} \right\rangle}_{I_1}$$

$$+ \frac{\rho}{2}\left\| \boldsymbol{L}^{l+1} - \begin{bmatrix} \boldsymbol{I}_r & -\boldsymbol{B}^l \\ -\left(\boldsymbol{B}^l\right)^\top & \operatorname{Diag}\left(\left(\boldsymbol{B}^l\right)^\top\mathbf{1}_r\right) \end{bmatrix} \right\|_{\mathrm{F}}^2 - \frac{\rho}{2}\left\| \boldsymbol{L}^{l+1} - \begin{bmatrix} \boldsymbol{I}_r & -\boldsymbol{B}^{l+1} \\ -\left(\boldsymbol{B}^{l+1}\right)^\top & \operatorname{Diag}\left(\left(\boldsymbol{B}^{l+1}\right)^\top\mathbf{1}_r\right) \end{bmatrix} \right\|_{\mathrm{F}}^2$$

$$+ \frac{p+\nu}{n}\sum_{i=1}^{n}\log\left(1 + \frac{h_i + \operatorname{tr}\left(\boldsymbol{B}^l\boldsymbol{G}_i\right)}{\nu}\right) - \frac{p+\nu}{n}\sum_{i=1}^{n}\log\left(1 + \frac{h_i + \operatorname{tr}\left(\boldsymbol{B}^{l+1}\boldsymbol{G}_i\right)}{\nu}\right). \tag{34}$$

For the term $I_1$, we have

$$I_1 = \left\langle \begin{bmatrix} \boldsymbol{I}_r & -\boldsymbol{B}^{l+1} \\ -\left(\boldsymbol{B}^{l+1}\right)^\top & \operatorname{Diag}\left(\left(\boldsymbol{B}^{l+1}\right)^\top\mathbf{1}_r\right) \end{bmatrix} - \begin{bmatrix} \boldsymbol{I}_r & -\boldsymbol{B}^l \\ -\left(\boldsymbol{B}^l\right)^\top & \operatorname{Diag}\left(\left(\boldsymbol{B}^l\right)^\top\mathbf{1}_r\right) \end{bmatrix}, \boldsymbol{Y}^l \right\rangle$$

$$- \rho\left\| \boldsymbol{L}^{l+1} - \begin{bmatrix} \boldsymbol{I}_r & -\boldsymbol{B}^{l+1} \\ -\left(\boldsymbol{B}^{l+1}\right)^\top & \operatorname{Diag}\left(\left(\boldsymbol{B}^{l+1}\right)^\top\mathbf{1}_r\right) \end{bmatrix} \right\|_{\mathrm{F}}^2, \tag{35}$$

where the equality follows from the updating of $\boldsymbol{Y}^{l+1}$ as below

$$\boldsymbol{Y}^{l+1} = \boldsymbol{Y}^l - \rho\left( \boldsymbol{L}^{l+1} - \begin{bmatrix} \boldsymbol{I}_r & -\boldsymbol{B}^{l+1} \\ -\left(\boldsymbol{B}^{l+1}\right)^\top & \operatorname{Diag}\left(\left(\boldsymbol{B}^{l+1}\right)^\top\mathbf{1}_r\right) \end{bmatrix} \right). \tag{36}$$

Recall that $\boldsymbol{B}^{l+1}$ is a stationary point of the problem

$$\underset{\boldsymbol{B} \geq \boldsymbol{0}, \boldsymbol{B} \boldsymbol{1} = \boldsymbol{1}}{\text{minimize}} \frac{p+\nu}{n} \sum_{i=1}^{n} \log \left(1 + \frac{h_i + \text{tr}\left(\boldsymbol{B} \boldsymbol{G}_i\right)}{\nu}\right) + \frac{\rho}{2} \left\| \boldsymbol{L}^{l+1} - \begin{bmatrix} \boldsymbol{I}_r & -\boldsymbol{B} \\ -\boldsymbol{B}^{\top} & \text{Diag}\left(\boldsymbol{B}^{\top} \boldsymbol{1}_r\right) \end{bmatrix} \right\|_{\text{F}}^{2}$$

$$- \left\langle \begin{bmatrix} \boldsymbol{I}_r & -\boldsymbol{B} \\ -\boldsymbol{B}^{\top} & \text{Diag}\left(\boldsymbol{B}^{\top} \boldsymbol{1}_r\right) \end{bmatrix}, \boldsymbol{Y}^l \right\rangle, \quad (37)$$

The set of stationary points for the optimization (37) is defined by

$$\mathcal{X} = \left\{ \boldsymbol{B} \mid \langle \nabla g_l(\boldsymbol{B}), \boldsymbol{Z} - \boldsymbol{B} \rangle \geq 0, \forall \boldsymbol{Z} \geq \boldsymbol{0}, \boldsymbol{Z} \boldsymbol{1} = \boldsymbol{1} \right\}, \quad (38)$$

where $g_l(\boldsymbol{w})$ is the objective function in (37). By taking $\boldsymbol{Z} = \boldsymbol{B}^l$ and $\boldsymbol{B} = \boldsymbol{B}^{l+1}$ in (38), we obtain

$$\langle \nabla g_l(\boldsymbol{B}^{l+1}), \boldsymbol{B}^l - \boldsymbol{B}^{l+1} \rangle = \langle \nabla h(\boldsymbol{B}^{l+1}), \boldsymbol{B}^l - \boldsymbol{B}^{l+1} \rangle + \rho \langle 2\boldsymbol{B}^{l+1} + \boldsymbol{1}_r \boldsymbol{1}_r^{\top} \boldsymbol{B}^{l+1}, \boldsymbol{B}^l - \boldsymbol{B}^{l+1} \rangle$$

$$+ \left\langle \begin{bmatrix} \boldsymbol{I}_r & -\boldsymbol{B}^{l+1} \\ -\left(\boldsymbol{B}^{l+1}\right)^{\top} & \text{Diag}\left(\left(\boldsymbol{B}^{l+1}\right)^{\top} \boldsymbol{1}_r\right) \end{bmatrix} - \begin{bmatrix} \boldsymbol{I}_r & -\boldsymbol{B}^{l} \\ -\left(\boldsymbol{B}^{l}\right)^{\top} & \text{Diag}\left(\left(\boldsymbol{B}^{l}\right)^{\top} \boldsymbol{1}_r\right) \end{bmatrix}, \boldsymbol{Y}^l + \rho \boldsymbol{L}^{l+1} \right\rangle \geq 0, \quad (39)$$

where $h(\boldsymbol{B}) := \frac{p+\nu}{n} \sum_{i=1}^{n} \log \left(1 + \frac{h_i + \text{tr}\left(\boldsymbol{B} \boldsymbol{G}_i\right)}{\nu}\right)$.

Substituting (35) and (39) into (34), we obtain

$$L_\rho \left(\boldsymbol{L}^{l+1}, \boldsymbol{B}^l, \boldsymbol{Y}^l\right) - L_\rho \left(\boldsymbol{L}^{l+1}, \boldsymbol{B}^{l+1}, \boldsymbol{Y}^{l+1}\right) \geq \frac{\rho}{2} \left\| \tilde{\boldsymbol{B}}^{l+1} - \tilde{\boldsymbol{B}}^l \right\|_{\text{F}}^2$$

$$- \frac{L_h}{2} \left\| \boldsymbol{B}^{l+1} - \boldsymbol{B}^l \right\|_{\text{F}}^2 - \frac{1}{\rho} \left\| \boldsymbol{Y}^{l+1} - \boldsymbol{Y}^l \right\|_{\text{F}}^2, \quad (40)$$

where $\tilde{\boldsymbol{B}}^l := \begin{bmatrix} \boldsymbol{I}_r & -\boldsymbol{B}^l \\ -\left(\boldsymbol{B}^l\right)^{\top} & \text{Diag}\left(\left(\boldsymbol{B}^l\right)^{\top} \boldsymbol{1}_r\right) \end{bmatrix}$, and the inequality follows from the fact that $h(\boldsymbol{B})$ is a concave function and has $L_h$-Lipschitz continuous gradient where $L_h > 0$ is a constant, and thus we obtain

$$h\left(\boldsymbol{B}^l\right) - h\left(\boldsymbol{B}^{l+1}\right) - \left\langle \nabla h\left(\boldsymbol{B}^{l+1}\right), \boldsymbol{B}^l - \boldsymbol{B}^{l+1} \right\rangle \geq -\frac{L_h}{2} \left\| \boldsymbol{B}^{l+1} - \boldsymbol{B}^l \right\|_{\text{F}}^2. \quad (41)$$

By calculation, we obtain that if $\rho$ is sufficiently large such that

$$\rho \geq \max \left(L_h, \max_l \frac{2 \left\| \boldsymbol{Y}^{l+1} - \boldsymbol{Y}^l \right\|_{\text{F}}}{\left\| \boldsymbol{B}^{l+1} - \boldsymbol{B}^l \right\|_{\text{F}}}\right), \quad (42)$$

then we have

$$L_\rho \left(\boldsymbol{L}^{l+1}, \boldsymbol{B}^l, \boldsymbol{Y}^l\right) - L_\rho \left(\boldsymbol{L}^{l+1}, \boldsymbol{B}^{l+1}, \boldsymbol{Y}^{l+1}\right) \geq 0. \quad (43)$$

Together with (33) and (43), we conclude that

$$L_\rho \left(\boldsymbol{L}^l, \boldsymbol{B}^l, \boldsymbol{Y}^l\right) \geq L_\rho \left(\boldsymbol{L}^{l+1}, \boldsymbol{B}^l, \boldsymbol{Y}^l\right) \geq L_\rho \left(\boldsymbol{L}^{l+1}, \boldsymbol{B}^{l+1}, \boldsymbol{Y}^{l+1}\right), \quad (44)$$

for any $l \in \mathbb{N}_+$. ■

Now we are ready to prove Theorem 2. By Lemma 5, the sequence $\left\{\left(\boldsymbol{L}^l, \boldsymbol{B}^l, \boldsymbol{Y}^l\right)\right\}$ is bounded. Therefore, there exists at least one convergent subsequence $\left\{\left(\boldsymbol{L}^{l_s}, \boldsymbol{B}^{l_s}, \boldsymbol{Y}^{l_s}\right)\right\}_{s \in \mathbb{N}}$, which converges to a limit point denoted by $\left\{\left(\boldsymbol{L}^{l\infty}, \boldsymbol{B}^{l\infty}, \boldsymbol{Y}^{l\infty}\right)\right\}$. By Lemma 6, we obtain that $L_\rho \left(\boldsymbol{L}^l, \boldsymbol{B}^l, \boldsymbol{Y}^l\right)$ is monotonically decreasing and lower bounded, and thus is convergent. Note that the function $\log \det^*(\boldsymbol{\Theta})$ is continuous over the set of $p$-dimensional positive semidefinite matrices of rank $p - k$, i.e., $\left\{\boldsymbol{L} \in \mathbb{S}_+^p \mid \text{rank}(\boldsymbol{L}) = p - k\right\}$.

We can then obtain

$$\lim_{l \to +\infty} L_\rho \left(\boldsymbol{L}^l, \boldsymbol{B}^l, \boldsymbol{Y}^l\right) = L_\rho \left(\boldsymbol{L}^\infty, \boldsymbol{B}^\infty, \boldsymbol{Y}^\infty\right) = L_\rho \left(\boldsymbol{L}^{l\infty}, \boldsymbol{B}^{l\infty}, \boldsymbol{Y}^{l\infty}\right).$$

Then, (40), (42) and (44) together yields

$$L_\rho(\boldsymbol{L}^l, \boldsymbol{B}^l, \boldsymbol{Y}^l) - L_\rho(\boldsymbol{L}^{l+1}, \boldsymbol{B}^{l+1}, \boldsymbol{Y}^{l+1}) \geq \rho \left\| \boldsymbol{L}^{l+1} - \begin{bmatrix} \boldsymbol{I}_r & -\boldsymbol{B}^{l+1} \\ -\left(\boldsymbol{B}^{l+1}\right)^\top & \mathsf{Diag}\left(\left(\boldsymbol{B}^{l+1}\right)^\top \boldsymbol{1}_r\right) \end{bmatrix} \right\|_{\mathrm{F}}^2 . \tag{45}$$

Thus, we obtain

$$\lim_{l \to +\infty} \left\| \boldsymbol{L}^l - \begin{bmatrix} \boldsymbol{I}_r & -\boldsymbol{B}^l \\ -\left(\boldsymbol{B}^l\right)^\top & \mathsf{Diag}\left(\left(\boldsymbol{B}^l\right)^\top \boldsymbol{1}_r\right) \end{bmatrix} \right\|_{\mathrm{F}} = 0. \tag{46}$$

Obviously, $\left\| \boldsymbol{L}^{l_s} - \begin{bmatrix} \boldsymbol{I}_r & -\boldsymbol{B}^{l_s} \\ -\left(\boldsymbol{B}^{l_s}\right)^\top & \mathsf{Diag}\left(\left(\boldsymbol{B}^{l_s}\right)^\top \boldsymbol{1}_r\right) \end{bmatrix} \right\|_{\mathrm{F}} \to 0$ also hold for any subsequence as $s \to +\infty$, which implies that $\boldsymbol{Y}^{l\infty}$ satisfies the condition of the stationary point of $L_\rho(\boldsymbol{L}, \boldsymbol{B}, \boldsymbol{Y})$ with respect to $\boldsymbol{Y}$. Following from (36), we obtain

$$\lim_{l \to +\infty} \left\| \boldsymbol{Y}^{l+1} - \boldsymbol{Y}^l \right\|_{\mathrm{F}} = 0. \tag{47}$$

Together with (40), we obtain

$$\lim_{l \to +\infty} \left\| \boldsymbol{B}^{l+1} - \boldsymbol{B}^l \right\|_{\mathrm{F}} = 0. \tag{48}$$

Let $\left\{ \left( \boldsymbol{L}^{l\infty}, \boldsymbol{B}^{l\infty}, \boldsymbol{Y}^{l\infty} \right) \right\}$ be the limit point of any subsequence $\left\{ \left( \boldsymbol{L}^{l_s}, \boldsymbol{B}^{l_s}, \boldsymbol{Y}^{l_s} \right) \right\}_{s \in \mathbb{N}}$. Following from (47) and (48), we obtain that $\boldsymbol{L}^{l\infty}$ minimizes the following subproblem

$$\boldsymbol{L}^{l\infty} = \underset{\substack{\mathsf{rank}(\boldsymbol{L}) = p-k \\ \boldsymbol{L} \succeq \boldsymbol{0}}}{\arg\min} -\log \det{}^*(\boldsymbol{L}) + \langle \boldsymbol{L}, \boldsymbol{Y}^{l\infty} \rangle + \frac{\rho}{2} \left\| \boldsymbol{L} - \begin{bmatrix} \boldsymbol{I}_r & -\boldsymbol{B}^{l\infty} \\ -\left(\boldsymbol{B}^{l\infty}\right)^\top & \mathsf{Diag}\left(\left(\boldsymbol{B}^{l\infty}\right)^\top \boldsymbol{1}_r\right) \end{bmatrix} \right\|_{\mathrm{F}}^2 .$$

Therefore, we conclude that $\boldsymbol{L}^{l\infty}$ satisfies the condition of stationary point of $L_\rho(\boldsymbol{L}, \boldsymbol{B}, \boldsymbol{Y})$ with respect to $\boldsymbol{L}$. Similarly, $\boldsymbol{B}^{l\infty}$ is the stationary point of the following problem

$$\underset{\boldsymbol{B} \geq \boldsymbol{0}, \boldsymbol{B1} = \boldsymbol{1}}{\mathsf{minimize}} \frac{p+\nu}{n} \sum_{i=1}^n \log\left(1 + \frac{h_i + \mathsf{tr}\left(\boldsymbol{BG}_i\right)}{\nu}\right) + \frac{\rho}{2} \left\| \boldsymbol{L}^{l\infty} - \begin{bmatrix} \boldsymbol{I}_r & -\boldsymbol{B} \\ -\boldsymbol{B}^\top & \mathsf{Diag}\left(\boldsymbol{B}^\top \boldsymbol{1}_r\right) \end{bmatrix} \right\|_{\mathrm{F}}^2$$
$$- \left\langle \begin{bmatrix} \boldsymbol{I}_r & -\boldsymbol{B} \\ -\boldsymbol{B}^\top & \mathsf{Diag}\left(\boldsymbol{B}^\top \boldsymbol{1}_r\right) \end{bmatrix}, \boldsymbol{Y}^{l\infty} \right\rangle .$$

As a result, $\boldsymbol{B}^{l\infty}$ satisfies the condition of stationary point of $L_\rho(\boldsymbol{L}, \boldsymbol{B}, \boldsymbol{Y})$ with respect to $\boldsymbol{B}$.

To sum up, we can conclude that any limit point $\left\{ \left( \boldsymbol{L}^{l\infty}, \boldsymbol{B}^{l\infty}, \boldsymbol{Y}^{l\infty} \right) \right\}$ of the sequence generated by Algorithm 3 is a stationary point of $L_\rho(\boldsymbol{L}, \boldsymbol{B}, \boldsymbol{Y})$. ∎

# E A PGD Algorithm for Subproblem (24)

One of the steps of Algorithm 3 involves solving a strongly convex problem described in Subproblem (24) in the main manuscript. While using convex solvers is a viable alternative, it may be more efficient to use a custom algorithm. Here we present a simple yet efficient PGD algorithm to solve Subproblem (24).

Let $g(\boldsymbol{B}) \triangleq \mathsf{tr}\left(\boldsymbol{B}\left(\boldsymbol{H} + \boldsymbol{M}^j\right)\right) + \rho \|\boldsymbol{B}\|_{\mathrm{F}}^2 + \frac{\rho}{2} \boldsymbol{1}_r^\top \boldsymbol{BB}^\top \boldsymbol{1}_r$ be the objective function of Subproblem (24), its gradient is given as $\nabla g(\boldsymbol{B}) = \left(\boldsymbol{H} + \boldsymbol{M}^j\right)^\top + \rho\left(2\boldsymbol{I}_r + \boldsymbol{11}^\top\right)\boldsymbol{B}$. Then, the PGD iterates can be written as

$$\boldsymbol{B}^{j+1} = \underset{\boldsymbol{B} \geq \boldsymbol{0}, \boldsymbol{B1}_q = \boldsymbol{1}_r}{\arg\min} \left\| \boldsymbol{B} - \left(\boldsymbol{B}^j - \alpha_j \nabla g(\boldsymbol{B}^j)\right) \right\|_{\mathrm{F}}^2, \tag{49}$$

where $\alpha_j$ is the learning rate, which can be updated using backtracking line search rules such as the one presented in (11) in the main manuscript. Problem (49) is an Euclidean projection of the rows of $\boldsymbol{B}^j - \alpha_j \nabla g(\boldsymbol{B}^j)$ onto the probability simplex. The unique solution to Problem (49) can be found

efficiently via several algorithms [35–37] whose theoretical worst-case complexity is $O(rq^2)$ but the observed practical complexity is $O(rq)$ [38].

Since the objective function of Subproblem (24) is strongly convex and its feasible set is compact, the PGD iterates converge to the global solution of Subproblem (24) [32].