# OpenReview forum: "Learning Bipartite Graphs: Heavy Tails and Multiple Components"
_NeurIPS.cc/2022/Conference — NeurIPS 2022 Accept_

### Official Review · Reviewer_Jahb · 2022-07-09

**Rating:** 5
**Confidence:** 4
**Soundness:** 3 good
**Presentation:** 3 good
**Contribution:** 3 good

**Summary:**

This paper investigates the problem of learning undirected, bipartite graphs under different statistical models. The authors propose an optimization framework based on projected gradient descent for learning bipartite graphs under the Gaussian Markov random field model and the problem formulation is extended to learning bipartite graphs under multi-variate Student-t distribution and under multi-component assumption. The key feature of different algorithms proposed is leveraging the properties of bipartite graphs with statistical properties of the considered data models to efficiently infer graph structures. Experiments on a real world dataset demonstrate that the proposed algorithms significantly outperform the state of the art on a multi-component clustering task.

**Questions:**

1. Please provide a reference to the exact lemma for block matrices from [24].
2. The definition of operator $P_{\Delta}$ seems to be missing.
3. I did not follow how the problem in (10) amounts to a Euclidean projection on the probability simplex. I suggest adding more details to accompany the following statement after (10): Problem (10) is a Euclidean projection of the rows....
4. Please provide mathematical definitions for modularity and accuracy in Section 4.
5. I believe that the paper will benefit from a more elaborate empirical evaluation focused on learning of bipartite graphs (perhaps on a synthetic dataset) to conclusively demonstrate the utility of the proposed algorithms and any gains over state of the art approaches.

**Limitations:**

Yes.

**Strengths And Weaknesses:**

Strengths:
1. Structure learning of bipartite graphs is a well-motivated problem. Learning bipartite graphs under Gaussian model is well-defined and uses standard technical arguments. The main contributions of interest are learning bipartite graphs under the Student-t distribution to accommodate data models with outliers or heavy-tailed distribution. For this purpose, the authors provide an iterative algorithm based on majorization-minimization framework and extend it to the learning of $k$-component, bipartite graphs.
2. The mathematical arguments are sound and the rationale behind various choices in the implementations are (usually) well described.
3. The paper is clearly written and easy to follow.

Weaknesses:
I find the experiments section to be too narrowly focused and possible inadequate to demonstrate learning of bipartite graphs.
Specifically, the experiments focus only on a clustering task and ignore learning of edge weights in these graphs (note that here I refer to the edge weights within the identified clusters as well). The two metrics that quantify accuracy of clustering and the modularity of identified clusters seem to be insufficient in this regard.

---

> ### Author Response · Authors · 2022-07-30
> **Answers to General Comments**
>
> Dear Reviewer,
>
> > Strengths:
>
> > 1. Structure learning of bipartite graphs is a well-motivated problem. Learning bipartite graphs under Gaussian model is well-defined and uses standard technical arguments. The main contributions of interest are learning bipartite graphs under the Student-t distribution to accommodate data models with outliers or heavy-tailed distribution. For this purpose, the authors provide an iterative algorithm based on majorization-minimization framework and extend it to the learning of k-component, bipartite graphs.
>
> We would like to thank you for recognising the importance and motivation behind problem being tackled as well as our efforts
> in proposing an algorithmic solution for it.
>
> > 2. The mathematical arguments are sound and the rationale behind various choices in the implementations are (usually) well described.
>
> We would like to thank you for appreciating the soundness of the proposed method as well as the implementation decision required.
>
> > 3. The paper is clearly written and easy to follow.
>
> We are happy to know that our efforts into making the manuscript concise and clear is appreciated by the reviewer.
>
> > Weaknesses: I find the experiments section to be too narrowly focused and possible inadequate to demonstrate learning of bipartite graphs.
> Specifically, the experiments focus only on a clustering task and ignore learning of edge weights in these graphs (note that here I refer
> to the edge weights within the identified clusters as well). The two metrics that quantify accuracy of clustering and the modularity of
> identified clusters seem to be insufficient in this regard.
>
> We thank the reviewer for expressing their concern regarding the experiments being too narrowly focused on the clustering task
> and the edge weights were ignored. We would like to clarify that, while edge weights are ignored for the computation of accuracy, the same
> is not true for the value of modularity, since the modularity is a function of the adjacency matrix [35]. Therefore, edge weights play a
> crucial role in the performance evaluation carried out in our work.

---

> ### Author Response · Authors · 2022-07-30
> **Answers to Questions**
>
> > Please provide a reference to the exact lemma for block matrices from [24].
>
> Thank you for suggesting more clarifications on that reference.
> The lemma can be found in page 4 of [24]. Therefore, we will cite it as [24, pp. 4].
>
> > The definition of $P_\Delta$ operator seems to be missing.
>
> Thank you for pointing that out. We will explicitly mention that $P_\Delta(M)$ is defined as the Euclidean projection of
> the rows of $M$ onto the probability simplex.
>
> > I did not follow how the problem in (10) amounts to a Euclidean projection on the probability simplex. I suggest adding more details to accompany the following statement after (10): Problem (10) is a Euclidean projection of the rows....
>
> Thank you for raising that this part needs more clarification.
> We will add that, since the constraints of Problem (10) can be decoupled for each row, the problem can be solved row-by-row,
> and for each row, the problem amounts to an Euclidean projection onto the probability simplex.
>
> > Please provide mathematical definitions for modularity and accuracy in Section 4.
>
> Thank you for this suggestion. For accuracy, we mention in words in the beginning of Section 4.
> As for modularity, we did not add its expression due to page limit, but we stated a reference ([35])
> where its expression can be easily found. However, we will add the mathematical expressions of both accuracy and modularity
> to the supplementary material. Thank you for this suggestion again.
>
> > I believe that the paper will benefit from a more elaborate empirical evaluation focused on learning of bipartite graphs (perhaps on a synthetic dataset) to conclusively demonstrate the utility of the proposed algorithms and any gains over state of the art approaches.
>
> We thank the reviewer for raising their concern that a more elaborate empirical evaluation would demonstrate the
> utility of the proposed algorithm over state-of-the-art methods.
> While we agree that more experiments are always better, we have to be selective due to page limit.
> We designed experiments that show, quite evidently, the advantages and robustness of our methods over state-of-the-art approaches.
> For example, in Figure 3, we can clearly observe the shortcomings of the SGLA [11] method that allows the existence of isolated nodes,
> whereas our method overcomes that by using linear constraints on the node degrees. Still in Figure 3, our method (Figure 3a) outperforms
> SOGB [9] (Figure 3b) by a significant margin and we attribute that outperformance to the heavy-tailed assumption that our method employs.
>
> While we agree that an experiment with synthetic data would be insightful, we decided to focus on real datasets due to their practical
> importance in real life problems as well as page limit of the manuscript.  In addition, we would like to mention that the financial
> datasets fit well the theoretical assumptions of the paper, i.e.,
> the heavy-tail characteristic of stocks returns is well-known in the literature. In addition, we
> performed experiments in a rolling window basis, which let us evaluate the performance of the proposed methods over many different chunks
> of the data, further providing evidence of their outperformance. In the supplementary material, we provided additional experiments
> showcasing the empirical convergence of the proposed algorithms as well as an experiment with 9 sectors and 362 stocks of the SP500 over
> 10 years of data.
>
> Finally, we would like to express our gratitude to the reviewer for taking time to review our manuscript, the supportive comments, and
> the insightful suggestions. We hope we answered the reviewer's concerns appropriately and, if that is the case, we would appreciate if
> our score could be reevaluated. Finally, we are happy to clarify any additional concerns if there are any.

---

### Official Review · Reviewer_C5Ut · 2022-07-10

**Rating:** 4
**Confidence:** 4
**Soundness:** 2 fair
**Presentation:** 2 fair
**Contribution:** 2 fair

**Summary:**

Paper proposes methods for graphical model selection from i.i.d. observations. The graph structure is constrainted to consist of compoeents being bipartite graphs with two node types (objects and classes). Two types of methods are proposed: one for Gaussian models and another one for student-t distributed data that is suited for data plagued by outliers.


**Questions:**

* What is "..a verifiably convergent iterative algorithm" ?

* What is a "..unconstrained Gaussian Markov random fields (MRF)" ?

* What do you mean by "..bipartite graphs under the MRF assumption." and " from data under probabilistic assumptions"? Do you mean that the underlying graph structure encodes pairwise local, global Markov properties ? (https://en.wikipedia.org/wiki/Markov_random_field)  It would also be useful to write down the prob. distribution of the observed data more explicitly.

* What do you mean by "..We design an efficient optimisation algorithm" ? Does your algorithm achieve fundamenal limits regarding number of observations or computational complexity ? In which sense is your algorithm "efficient" ?

* What is "a Laplacian-constrained precision matrix"?

* What is the relation between matrix A in (3) in the Laplacian matrix that you are after ?

* Could you explain in more detail what you mean by "..our proposed methods and algorithms"

* What is an "observed practical complexity" ?

* What are "heavy-tailed events" and "heavy-tailed bipartite graphs"?

* Can you pls provide more details for the arguing "...nonconvex due to the summation over concave terms and hence it is difficult to be dealt with directly."

* How does the rank constraint in (18) enter into (19)?

* What are the "node labels" that you refer to in the Experiments  ?

* What is the precise definition for the accuracy and modularity depicted in Figure 1 ?

* Is Algorithm 3 returning a graph or a matrix ?

* You might put your approach into context of another family of networked-structured probabilistic models:

A. Jung, "Networked Exponential Families for Big Data Over Networks," in IEEE Access, vol. 8, pp. 202897-202909, 2020, doi: 10.1109/ACCESS.2020.3033817.

In contrast, to probabilistic graphical models (such as GMRF), networked exponential families model local datasets (at nodes) using separate probability spaces. In contrast, GMRF use a common probability space on which all node attributes/observations are defined.
*

**Ethics Review Area:**

["I don’t know"]

**Strengths And Weaknesses:**

Strengths:
* Learning network structure of data is an enabling task for many follow up applications such as federated learning.
* Methods are evaluated on "real-world" data.

Weaknesses:

* The novelty is somewhat limited as the proposed methods are combinations of established approaches to graphical model selection via penalaized maximum likelihood methods. The penalty terms are well motivated by existing literature on learning bipartite graphs.
The main technical result seems to be Theorem 2 which characterizes the convergence of Algorithm 3. What is the practical relevance of Theorem 2 ? How is it used/confirmed in the numerical experiments ?

* The underlying probabilistic model should be written out in more detail. In particular, are the samples assumed to be i.i.d.. or a stationary process which would be more suitable for the numerical experiments which use time series data.

* The numerical experiments should be expanded by considered synthetic datasets that allow to measure the estimation error of Algorithm 1 - 3. Moreover, it would be interesting to compare with recent methods for graphical model section from samples forming (non-)stationary processes:

A. Jung, G. Hannak and N. Goertz, "Graphical LASSO based Model Selection for Time Series," in IEEE Signal Processing Letters, vol. 22, no. 10, pp. 1781-1785, Oct. 2015, doi: 10.1109/LSP.2015.2425434.

A. Jung, "Learning the Conditional Independence Structure of Stationary Time Series: A Multitask Learning Approach," in IEEE Transactions on Signal Processing, vol. 63, no. 21, pp. 5677-5690, Nov.1, 2015, doi: 10.1109/TSP.2015.2460219.

N. Tran, O. Abramenko and A. Jung, "On the Sample Complexity of Graphical Model Selection From Non-Stationary Samples," in IEEE Transactions on Signal Processing, vol. 68, pp. 17-32, 2020, doi: 10.1109/TSP.2019.2956687.

* The paper would be much stronger if it would offer theoretical performance guarantees, e.g., required number of observations to ensure a prescribed estimation quality.

---

> ### Author Response · Authors · 2022-07-30
> **Answers to General Comments**
>
> > Strengths:
>
> > Learning network structure of data is an enabling task for many follow up applications such as federated learning.
> Methods are evaluated on "real-world" data.
>
> We appreciate that the reviewer expresses the importance of learning network structure of data and their follow up applications
> including federated learning. We also would like to thank the reviewer for mentioning as an strength that the methods are evaluated
> in real-world data.
>
> > Weaknesses:
>
> > The novelty is somewhat limited as the proposed methods are combinations of established approaches to graphical model selection
> via penalaized maximum likelihood methods. The penalty terms are well motivated by existing literature on learning bipartite graphs.
>
>
> We would like to thank the reviewer for expressing their concern regarding the limited novelty. We would like to mention that
> learning a bipartite graph from a Markov Random Field approach, while assuming a Student-t distribution for the data
> generated by the graph nodes, and extending that to learn a k-component bipartite graph, is a novel problem that requires nontrivial
> resources, like Theorem 2. Therefore, we respectfully disagree with the statement that the proposed methods are
> "combinations of established approaches to graphical model selection via penalaized  maximum likelihood methods".
>
> > The main technical result seems to be Theorem 2 which characterizes the convergence of Algorithm 3.
> What is the practical relevance of Theorem 2 ? How is it used/confirmed in the numerical experiments ?
>
> We would like to thank the reviewer for recognising our technical results.
> Theorem 2 acts as an insurance for the convergence of Algorithm 3. From a practical perspective,
> we performed empirical convergence experiments detailed in Section 1 of the Supplementary Material
> that showcase the practical convergence trend of the proposed algorithms including Algorithm 3.
> We also would like to add that it is crucial for any numerical algorithm, especially the ones dealing with nonconvex optimization problems,
> that experiments with different initial points are performed. On that front, Figure 2 in Section 1 of the Supplementary Material
> shows experimental results that verify the convergence of the proposed algorithms with different initial points.
>
>
> > The underlying probabilistic model should be written out in more detail. In particular, are the samples assumed to be i.i.d.. or a stationary process which would be more suitable for the numerical experiments which use time series data.
>
> We thank the reviewer for raising this concern. We would like to mention that the assumptions for data generating process as well as
> the structure of the Markov random field are presented in the beginning of Section 2.
> The proposed estimators are based on the maximum likelihood approach, hence the samples of the multivariate
> Student-t (or Gaussian for Algorithm 1) are assumed to be iid.
>
> > The numerical experiments should be expanded by considered synthetic datasets that allow to measure the estimation error of Algorithm 1 - 3. Moreover, it would be interesting to compare with recent methods for graphical model section from samples forming (non-)stationary processes:
>
> > A. Jung, G. Hannak and N. Goertz, "Graphical LASSO based Model Selection for Time Series," in IEEE Signal Processing Letters, Oct. 2015
>
> > A. Jung, "Learning the Conditional Independence Structure of Stationary Time Series: A Multitask Learning Approach," in IEEE Transactions on Signal Processing, Nov.1, 2015,
>
> > N. Tran, O. Abramenko and A. Jung, "On the Sample Complexity of Graphical Model Selection From Non-Stationary Samples," in IEEE Transactions on Signal Processing,  2020
>
> > The paper would be much stronger if it would offer theoretical performance guarantees, e.g., required number of observations to ensure a prescribed estimation quality.
>
> Firstly, thank you for pointing out those papers. We will be sure to mention and cite them in our manuscript.
>
> Secondly, we thank the reviewer for expressing their concern regarding the experimental section.
> While we agree that an experiment with synthetic data
> would be insightful, we decided to focus on real datasets due to their practical
> importance in real life problems as well as page limit of the
> manuscript.  In addition, we would like to mention that the financial datasets fit well the theoretical assumptions of the paper, i.e.,
> the heavy-tail characteristic of stocks returns is well-known in the literature. Moreover, we
> performed experiments in a rolling window basis, which let us evaluate the performance of the proposed methods over many different chunks of the data, further providing evidence of their outperformance. In the supplementary material, we provided additional experiments
> showcasing the empirical convergence of the proposed algorithms as well as an experiment with 9 sectors and 362 stocks of the SP500 over 10 years of data.

---

> ### Author Response · Authors · 2022-07-30
> **Answers to Questions Part 1**
>
> Firstly, we would like to thank the reviewer for asking clarifying questions.
>
> > What is "..a verifiably convergent iterative algorithm" ?
>
> It is a succinct, informal way to say that our algorithm generates a sequence of points that converge to a stationary point of the
> optimization problem and that this statement is proved theoretically.
>
> > What is a "..unconstrained Gaussian Markov random fields (MRF)" ?
>
> It is an GMRF whose constraints on its precision matrix are only that it lies in the positive-semidefinite cone.
>
> > What do you mean by "..bipartite graphs under the MRF assumption."
>
> "..bipartite graphs under the MRF assumption." is a Markov random field whose precision matrix is constrained to be a Laplacian matrix of a
> bipartite graph
>
> > " from data under probabilistic assumptions"? Do you mean that the underlying graph structure encodes pairwise local, global Markov properties ? (https://en.wikipedia.org/wiki/Markov_random_field)
>
> Yes, exactly.
>
> > What do you mean by "..We design an efficient optimisation algorithm" ? Does your algorithm achieve fundamenal limits regarding number of observations or computational complexity ? In which sense is your algorithm "efficient" ?
>
> Our algorithm is efficient from a theoretical and practical computational complexity.
> Our proposed methods in Algorithm 1 and 2 have complexity O(r*q^2),
> where usually r >> q (line 54), per iteration (line 129). The competing methods, i.e., SGA and SOBG (with k = 1) have per iteration
> complexity O(p^3), where p = r + q, since they rely on the eigen decomposition of the Adjacency (in case of SGA) or the Laplacian
> (in case of SOBG) matrices. In words, our method scales quadratically with the number of classes and linearly with the number of
> objects, whereas the competing methods scale cubically with the total number of nodes (i.e. objects + classes).
> The proposed method in Algorithm 3 and the competing methods, i.e., SGLA and SOBG (k > 1), all have the same theoretical
> computational complexity of O(p^3) as they rely on the eigendecomposition of the Laplacian and/or Adjacency matrix.
>
> > What is "a Laplacian-constrained precision matrix"?
>
> It's a precision matrix that satisfy Laplacian constraints, i.e., $\mathbf{L}1 = 0$, $L_{ij} = L_{ji} \leq 0$.
>
> > What is the relation between matrix A in (3) in the Laplacian matrix that you are after ?
>
> The expression in (3) is the optimization problem proposed by the authors in [9]. The matrix $A$ in (3)
> represents bipartite graph weights initially given (or obtained from, say, pairwise correlations).
> The matrix $A$ is used by the authors in [9] as an input data for them to obtained an improved version
> of the bipartite graph weights, through the solution of (3), which can be then used to construct the
> Laplacian matrix as expressed in equation (2).
>
> > Could you explain in more detail what you mean by "..our proposed methods and algorithms"
>
> In our manuscript we proposed three Algorithms for estimating bipartite graphs.
> Algorithm 1, for a simple connected bipartite graph under Gaussian assumptions.
> Algorithm 2, for a connected bipartite graph under Student-t assumptions.
> Algorithm 3, for a k-component bipartite graph under Student-t assumptions.
>
> > What is an "observed practical complexity" ?
>
> It's the computational time complexity actually measured after running the algorithm.
>
> > What are "heavy-tailed events" and "heavy-tailed bipartite graphs"?
>
> "heavy-tailed events" is a synonym for outliers. "heavy-tailed bipartite graphs" is a bipartite graph estimated under heavy-tailed
> probabilistic assumptions, such as Student-t.

---

> ### Author Response · Authors · 2022-07-30
> **Answers to Questions Part 2**
>
> > Can you pls provide more details for the arguing "...nonconvex due to the summation over concave terms and hence it is difficult to be dealt with directly."
>
> The logarithm function is concave. Sum of concave functions is also a concave function. That turns Problem (15) into a nonconvex problem.
> In general, nonconvex problems are difficult to deal with for a variety of reasons including because they may have many local minimas.
>
> > How does the rank constraint in (18) enter into (19)?
>
> It doesn't. The rank constraint on L is handled directly in the subproblem for L (equation (20)).
>
> > What are the "node labels" that you refer to in the Experiments ?
>
> In our experiments, the stocks are represented by the nodes of the graph. Therefore, the node labels correspond to the stock sectors that the nodes belong to. The stock sector are given by GICS (Global Industry Classification Standard).
>
> > What is the precise definition for the accuracy and modularity depicted in Figure 1 ?
>
> The definition of accuracy is described in line 207 in Section 4, i.e., "Accuracy is computed as the ratio
> between the number of correctly predicted node labels and the number of nodes in the objects set."
> Due to limited space, we did not present the mathematical expression for modularity, but it can be easily found
> in reference [35], which we mention explicitly in line 209 in Section 4. We will add that expression to our supplementary material.
>
> > Is Algorithm 3 returning a graph or a matrix ?
>
> Algorithm 3, as well as Algorithms 1 and 2, returns a Laplacian matrix, which uniquely represents a graph.
>
> > You might put your approach into context of another family of networked-structured probabilistic models:
>
> > A. Jung, "Networked Exponential Families for Big Data Over Networks," in IEEE Access, vol. 8, pp. 202897-202909, 2020, doi: 10.1109/ACCESS.2020.3033817.
>
> > In contrast, to probabilistic graphical models (such as GMRF), networked exponential families model local datasets (at nodes) using separate probability spaces. In contrast, GMRF use a common probability space on which all node attributes/observations are defined. *
>
> Thank you for the suggestion. Yes, indeed, there are different ways into looking at probabilistic models for networks.
>
> We would like to express our gratitude to the reviewer for taking time to review our manuscript, the supportive comments, and
> the insightful suggestions. We hope we answered the reviewer's concerns appropriately and, if that is the case, we would appreciate if
> our score could be reevaluated. Finally, we are happy to clarify any additional concerns if there are any.

---

### Official Review · Reviewer_LmQW · 2022-07-11

**Rating:** 7
**Confidence:** 3
**Soundness:** 2 fair
**Presentation:** 4 excellent
**Contribution:** 3 good

**Summary:**

The authors present a new optimization problem formulation for learning weighted bipartite graphs, and also extend the formulation to the presence of heavy tailed events (outliers) motivated by real-world problems. They propose a solution to both problem formulations based on PGD and MM, respectively. Experimental results on financial datasets demonstrate the superior performance of the approach compared to competing approaches, however the experiments themselves are limited.

**Questions:**

1) My major concern is that unfortunately the experimental results are focused on a single dataset and such have limited power to demonstrate the proposed approach. The experimental results section would benefit from a synthetic simulation in which the graphs are known and there are well defined outlier events Competing approaches (refs [9] and [11]) had much more detailed and varied experiments and the current paper would benefit from adding an additional synthetic experiment and real-world experiment.

2) The statement that there is a decline in accuracy / modularity surrounding an adverse event such as COVID 19 or the crash in 2008 (appendix) and this is a good property of the proposed method is unclear. How does this demonstrate that the approach can handle adverse effects if it results in worse performance? How in practice would this framework be used to identify such an event or perform under these heavy-tail conditions? In comparison, the competing approaches are unaffected or perform better after 2020.

3) What is the run-time complexity of this approach compared to competing methods (the supplement only details the proposed method)?


**Limitations:**

The authors did not describe limitations


**Strengths And Weaknesses:**

Originality:
The paper proposes a novel graph learning framework and efficient solutions in the convex and non-convex case.

Missing references from the graph learning literature:
* Dong, Xiaowen, et al. "Learning Laplacian matrix in smooth graph signal representations." IEEE Transactions on Signal Processing 64.23 (2016): 6160-6173.
* Kalofolias, Vassilis. "How to learn a graph from smooth signals." Artificial Intelligence and Statistics. PMLR, 2016
* Mateos, Gonzalo, et al. "Connecting the dots: Identifying network structure via graph signal processing." IEEE Signal Processing Magazine 36.3 (2019): 16-43.
* Hu, Chenhui, et al. "A spectral graph regression model for learning brain connectivity of Alzheimer’s disease." PloS one 10.5 (2015): e0128136.

Quality:
The paper is technically sound and the solutions appropriate. The experimental section is somewhat lacking: Further comments in the questions regarding the experiments.

Clarity:
The paper is well-written and the methods clearly laid out in algorithmic form for reproducibility.

Minor comments:
* Eq 3: V undefined
* Line 79: the statement that the method in [9] lacks statistical support seems somewhat harsh without toning down that it is motivated from a spectral graph theory perspective.
* Acronyms in section 4 are undefined (SOBG, SGA, SGLA)
* line 242: what does out of sample accuracy refer to? There was no reference to a train/test split with respect to this experiment so who are the out-of-samples? The new days?
* Figure 3 missing colorbar

---

> ### Author Response · Authors · 2022-07-28
> **Answers to General Comments**
>
> Dear Reviewer,
>
> > Originality: The paper proposes a novel graph learning framework and efficient solutions in the convex and non-convex case.
>
> Thank you for appreciating the originality of our work.
>
> > Missing references from the graph learning literature:
>
> > Dong, Xiaowen, et al. "Learning Laplacian matrix in smooth graph signal representations." IEEE Transactions on Signal Processing 64.23 (2016): 6160-6173.
>
> > Kalofolias, Vassilis. "How to learn a graph from smooth signals." Artificial Intelligence and Statistics. PMLR, 2016
>
> > Mateos, Gonzalo, et al. "Connecting the dots: Identifying network structure via graph signal processing." IEEE Signal Processing Magazine 36.3 (2019): 16-43.
>
> > Hu, Chenhui, et al. "A spectral graph regression model for learning brain connectivity of Alzheimer’s disease." PloS one 10.5 (2015): e0128136.
>
> Thanks for pointing out those references. We did not cite them previously mainly because of two reasons: they focus on the smooth-signal
> approach to graph learning, while we start off from the more statistically fundamented Markov Random Field approach; they do not focus on bipartite graphs, which is the motivation of our work. However, we do acknowledge they are key papers in the literature and we will cite them in the introduction.
>
> > Quality: The paper is technically sound and the solutions appropriate.
>
> Thank you for appreciating the technical quality and the solutions proposed in our paper.
>
> > Clarity: The paper is well-written and the methods clearly laid out in algorithmic form for reproducibility.
>
> We thank you very much for recognising the clarity of our manuscript especially the algorithmic forms that are
> definitely crucial for reproducibility. I just would like to mention that, in addition to that, the code to reproduce all the plots
> in our work is available in the supplementary material and it will be available in a GitHub repository in the future for the general public.
> We thank you very much again for appreciating our efforts on that front.
>
> > Eq 3: V undefined
>
> Thank you. We will mention the physical meaning of V, that is, the eigenvectors of the Laplacian matrix L.
>
> > Line 79: the statement that the method in [9] lacks statistical support seems somewhat harsh without toning down that it is motivated from a spectral graph theory perspective.
>
> We apologize that that statement came across as somewhat harsh, that was definitely not our intention. We will definitely rephrase it and mention that the method leverages spectral graph theory although no statistical distribution is prescribed.
>
> > Acronyms in section 4 are undefined (SOBG, SGA, SGLA)
>
> Thank you for pointing that out.
>
> > line 242: what does out of sample accuracy refer to? There was no reference to a train/test split with respect to this experiment so who are the out-of-samples? The new days?
>
> Please, disregard the word "out-of-sample". We apologize for the confusion, this was actually a typo.

---

> ### Author Response · Authors · 2022-07-28
> **Answers to Questions**
>
> > 1. My major concern is that unfortunately the experimental results are focused on a single dataset and such have limited power to demonstrate the proposed approach. The experimental results section would benefit from a synthetic simulation in which the graphs are known and there are well defined outlier events Competing approaches (refs [9] and [11]) had much more detailed and varied experiments and the current paper would benefit from adding an additional synthetic experiment and real-world experiment.
>
> We thank the reviewer for expressing their concern regarding the experimental section. While we agree that an experiment with synthetic data  would be insightful, we decided to focus on real datasets due to their practical importance in real life problems as well as page limit of the manuscript.  In addition, we would like to mention that the financial datasets fit well the theoretical assumptions of the paper, i.e.,  the heavy-tail characteristic of stocks returns is well-known in the literature. In order to avoid the issue of "single" dataset, we performed experiments in a rolling window basis, which let us evaluate the performance of the proposed methods over many different chunks  of the data, further providing evidence of their outperformance. In the supplementary material, we provided additional experiments  showcasing the empirical convergence of the proposed algorithms as well as an experiment with 9 sectors and 362 stocks of the SP500 over 10 years of data.
>
> > 2. The statement that there is a decline in accuracy / modularity surrounding an adverse event such as COVID 19 or the crash in 2008 (appendix) and this is a good property of the proposed method is unclear. How does this demonstrate that the approach can handle adverse effects if it results in worse performance? How in practice would this framework be used to identify such an event or perform under these heavy-tail conditions? In comparison, the competing approaches are unaffected or perform better after 2020.
>
> Thank you very much for raising this question. Let us start by mentioning that modularity is measure of how clustered together the  nodes in a graph are with respect to their ground-truth label (in our case, the label is given by the sector that the stock belongs to). The rationale  behind the sharp decline in modularity is as follows: during economic turmoils or crisis, it's common that there will be a panic sell-off  in the market, i.e., many investors look to sell their whole portfolio of stocks in order to avoid losses. This sell-off drives prices  of almost all the stocks down, which then creates correlations between stocks from seemingly distinct sectors. Those additional correlations between stocks (nodes) from different sectors (classes) reduce the value of modularity because now the nodes within each sector will be less
> clustered. We noticed that our method captures this natural behavior of the markets, therefore we conclude that our proposed method offers  a more realistic estimation of the network of stocks.  In practice, for example, we can use the modularity value of our proposed method to identify when a significant change in the market is happening, which could be helpful in other tasks such as risk management.
>
> > 3. What is the run-time complexity of this approach compared to competing methods (the supplement only details the proposed method)?
>
> We would like to thank the reviewer for raising this excellent question. We will separate our answer in two cases: (1) for connected bipartite graphs and (2) for k-component bipartite graphs. In the first scenario, from a theoretical perspective, our proposed methods in Algorithm 1 and 2 have complexity O(r*q^2), where usually r >> q (line 54), per iteration (line 129). The competing methods, i.e., SGA and SOBG (with k = 1) have per iteration  complexity O(p^3), where p = r + q, since they rely on the eigen decomposition of the Adjacency (in case of SGA) or the Laplacian  (in case of SOBG) matrices. In words, our method scales quadratically with the number of classes and linearly with the number of objects, whereas the competing methods scale cubically with the total number of nodes (i.e. objects + classes).
>
> In the second scenario, i.e., k-component bipartite graphs, the proposed method in Algorithm 3 and the competing methods, i.e., SGLA and SOBG (k > 1), all have the same theoretical computational complexity of O(p^3) as they rely on the eigendecomposition of the Laplacian and/or  Adjacency matrix.
>
> Finally, we would like to express our gratitude to the reviewer for taking time to review our manuscript, the supportive comments, and
> the insightful suggestions. We hope we answered the reviewer's concerns appropriately and, if that is the case, we would appreciate if
> our score could be reevaluated. Lastly, we are happy to clarify any additional concerns if there are any.

---

> > ### Comment · Reviewer_LmQW · 2022-08-09
> > **reply to authors**
> >
> > thanks for the detailed replies and clarifications. I recommend including these explanations in the revised paper. I am also satisfied with the additional experiments and raise my score to a 7.

---

> > > ### Author Response · Authors · 2022-08-09
> > > **Reply to reviewer**
> > >
> > > We would like to thank the reviewer very much for their feedback as well as their acknowledgement of the contributions of our manuscript.
> > >
> > > Thank you, the authors.

---

### Author Response · Authors · 2022-08-02
**Rebuttal Phase**

Dear reviewers,

Firstly, thank you very much for your comments.
Secondly, please find bellow our detailed responses to all your comments and questions.

We strived to answer your comments and questions in a clear and concise manner.
We hope those answers clarify the confusions that you may have had.
In addition, we would highly appreciate if that would lead to a change in score that better reflects the current state of the paper.
In general, we noticed that we received very good comments and scores about the presentation and significance of the contribution, but that is not reflected in the final rating.

Finally, please, do let us know if there are more questions or comments from your side, we are more than happy to answer them.

Best,
the authors.

---

### Author Response · Authors · 2022-08-07
**Additional Experimental Results on MNIST Data and Synthetic Data**

In order to fulfil the reviewer's request regarding experimental results, we performed two additional experiments, whose results will be added to the Supplementary Material.  The first one involving MNIST digits image data and the second one with synthetic data. We urge the reviewers to reconsider their rating score should the experiments below provide a more clear picture for the concerns they raised.

MNIST Experiment
----------------------

We conduct a soft-clustering experiment using MNIST image data.
MNIST provides a collection of $28\times 28$ grey-scaled images of hand-written digits
from $0$ to $9$.
The nodes in the classes set are defined such that every node corresponds to a unique digit.

We assign 500 nodes for the object set, each of which representing an image randomly selected from the testing set.
We randomly select the images in a stratified way, such that every label (digit) appears 50 times.
The signal for a node $v_i \in \mathcal{V}_q$, associated to digit $i$, is constructed by averaging
1000 randomly selected images from the training set whose label corresponds to $i$. We construct the
data matrix $\mathbf{X}$ by vectorizing and stacking the images column-wise. The quantitative measures
are then computed as an average of 50 realizations of this randomized experiment.

We then proceed to learn graphs by the proposed algorithms and benchmarks.
Similarly to the previous experiment, the final label assigned to the $i$-th image in the objects set
corresponds to $argmax_{j \in 1, \dots, q} B_{ij} - 1$.

The table below provides the quantitative results in terms of accuracy and modularity of the estimated graphs.
We observe that $\mathsf{GBG}$ (proposed) is the most accurate among the methods followed
by $\mathsf{SOBG}$, while $\mathsf{SGA}$ is not as competitive in terms of accuracy.

| Algorithm | Accuracy |  Modularity |
| --- | ----------- | -- |
| $\mathsf{GBG}$ (proposed) | **0.81**  | **0.39**|
| $\mathsf{SBG}$ (proposed) | 0.74  | 0.32 |
|  $\mathsf{SOBG}$ (benchmark) | 0.76 |0.29 |
| $\mathsf{SGA}$ (benchmark) | 0.62  | 0.35 |

Synthetic Data
-----------------

We generate synthetic data from a multivariate Gaussian distribution, where the covariance matrix of this multivariate Gaussian is set to be the pseudo-inverse of the Laplacian matrix of a bipartite graph (note that this setting is the same as that of the SGA/SGLA paper by Kumar et al. 2019).

We set the number of nodes of the bipartite graph to be $p = 510$, where $r=500$, and $q = 10$.
We then sample $n$ observations from this multivariate Gaussian, where we seek to measure the relative error (RE) between the true graph and the estimated ones.
We measure average RE for three sample size regimes: 1) small, (n=2), 2) medium (n=50), and large (n=500). The average RE is computed for 100 realizations in each scenario.

We observe from the table below that the proposed algorithms notably show a significant advantage for small and medium sample-size regimes. For large sample size, they perform statistically the same.

| Algorithm | n = 2 | n = 50 | n = 500 |
| --- | ----------- | -- | -- |
| $\mathsf{GBG}$ (proposed) | 0.69  | 0.34  | 0.11|
| $\mathsf{SBG}$ (proposed) | **0.68**  | **0.32**| **0.10** |
|  $\mathsf{SOBG}$ (benchmark) | 0.91 | 0.62 | 0.11 |
| $\mathsf{SGA}$ (benchmark) | 0.95  | 0.64  | 0.12 |

---

### Author Response · Authors · 2022-08-09
**Rebuttal Phase Feedback**

Dear reviewers,

Firstly, we would like to thank Reviewer LmQW for kindly getting back to us after our responses for their questions.

Secondly, we would like to ask Reviewers Jahb and C5Ut whether they have any more questions. We are happy to answer them.  We would also like to ask Reviewers Jahb and C5Ut whether they have any feedback on the responses that we elaborated for their questions.  We would really appreciate if our clarifying responses could result in a reevaluation of the score for our manuscript.

Thank you very much.

---

> ### Comment · Reviewer_Jahb · 2022-08-09
> **Thank you.**
>
> Thank you for your detailed responses and additional experiments. I will retain my original recommendation for this paper.

---

### Meta-Review · Area_Chair_9q5U · 2022-08-26

**Recommendation:** Accept
**Confidence:** Less certain

**Metareview:**

The reviewers agree that this work deals with an important problem and provided a sound solution.

**Award:**

No

---

### Decision · Program_Chairs · 2022-09-14

Accept